# Artesunate, EDTA, and colistin work synergistically against MCR-negative and -positive colistin-resistant *Salmonella*

**Yajun Zhai\*†, Peiyi Liu†, Xueqin Hu†, Changjian Fan, Xiaodie Cui, Qibiao He, Dandan He, Xiaoyuan Ma, Gongzheng Hu\***

Department of Pharmacology and Toxicology, College of Veterinary Medicine, Henan Agricultural University, Zhengzhou, China

## eLife Assessment

This **valuable** study addresses the growing threat of multidrug-resistant (MDR) pathogens by focusing on the enhanced efficacy of colistin when combined with artesunate and EDTA against colistin-resistant *Salmonella* strains. The evidence is **solid**, supported by comprehensive microbiological assays, molecular analyses, and in vivo experiments demonstrating the effectiveness of this synergic combination.

**\*For correspondence:**
yaolilab@126.com (GH);
zyj90518@126.com (YZ)

†These authors contributed equally to this work

**Competing interest:** See page

**Abstract** Discovering new strategies to combat the multidrug-resistant bacteria constitutes a major medical challenge of our time. Previously, artesunate (AS) has been reported to exert antibacterial enhancement activity in combination with β-lactam antibiotics via inhibition of the efflux pump AcrB. However, combination of AS and colistin (COL) revealed a weak synergistic effect against a limited number of strains, and few studies have further explored its possible mechanism of synergistic action. In this article, we found that AS and EDTA could strikingly enhance the antibacterial effects of COL against *mcr-1*⁻ and *mcr-1*⁺ *Salmonella* strains either in vitro or in vivo, when used in triple combination. The excellent bacteriostatic effect was primarily related to the increased cell membrane damage, accumulation of toxic compounds and inhibition of MCR-1. The potential binding sites of AS to MCR-1 (THR283, SER284, and TYR287) were critical for its inhibition of MCR-1 activity. Additionally, we also demonstrated that the CheA of chemosensory system and virulence-related protein SpvD were critical for the bacteriostatic synergistic effects of the triple combination. Selectively targeting CheA, SpvD, or MCR using the natural compound AS could be further investigated as an attractive strategy for the treatment of *Salmonella* infection. Collectively, our work opens new avenues toward the potentiation of COL and reveals an alternative drug combination strategy to overcome COL-resistant bacterial infections.

## Introduction

*Salmonella* is globally recognized as a major zoonotic foodborne pathogen that is responsible for food poisoning, gastroenteritis, and even life-threatening in animals and humans (*Galán-Relaño et al., 2023*; *Gangathraprabhu et al., 2020*). Antibiotics are commonly used to shorten the duration of illness and reduce infectivity. However, with the increasing use of antibiotics, the infections caused by multi-drug-resistant (MDR) pathogens, especially the carbapenemase-producing *Enterobacteriaceae*, have become a major source of public health concern (*Foletto et al., 2021*). The shortage of new antibiotics for these MDR bacteria strains has led to the reuse of polymyxins as the 'last resort' antimicrobial drug with the inevitable risk of emerging resistance (*Falagas and Kasiakou, 2005*).

Colistin (polymyxin E, COL), a fatty acyl oligopeptide antibiotic, is an active agent against Gram-negative ($G^-$) pathogens, and has been widely used to combat *Salmonella* infections. Generally, COL kills bacteria through a detergent-like effect, which the polycationic ring of COL electrostatically interacts with the cell envelope components, causing the competitive displacement of divalent cations calcium ($Ca^{2+}$) and magnesium ($Mg^{2+}$), destabilizing the membrane, thus killing the bacterium via the 'self-promoted uptake' pathway (**Kaye et al., 2016**). Beyond that, other models for the antibacterial activity have been reported, including vesicle–vesicle contact, hydroxyl radical death, inhibition of respiratory enzymes, and anti-endotoxin COL activity pathways (**El Sayed Ahmed et al., 2020**). Until now, numerous chromosomally or plasmid-mediated mechanisms underlying polymyxins resistance in $G^-$ bacteria have been identified, including intrinsic, mutation (e.g., PmrAB, PhoPQ, or AcrAB-TolC mutants), adaptation mechanisms, or horizontally acquired resistance via the phosphoethanolamine (pEtN) transferase genes *mcr-1–9* (**Carroll et al., 2019**; **Lima et al., 2018**; **Poirel et al., 2017**).

To tackle the increasing emergence of MDR pathogens, many alternative therapies, less costly and time-consuming than drug discovery, have been new areas of current research interest, involving the combination therapy of existing agents, the drug discovery from natural products, and the evaluation of drug resistance reversers (**Rosenthal, 2003**). A variety of antibiotic adjuvants that may or may not have direct antibacterial effects have been widely investigated to increase the effectiveness of current antibiotics or delay the emergence of drug resistance, such as β-lactamase inhibitors, aminoglycoside-modifying enzyme inhibitors, membrane permeabilizers, and efflux pump inhibitors (**Laws et al., 2019**). Several promising inhibitors, for example, zidebactam and pyrazolopyrimidine compounds, have been described as the β-lactam and aminoglycoside enhancers against $G^-$ bacteria (**Moya et al., 2017**; **Stogios et al., 2013**). Alternatively, numerous synthetic antimicrobial peptides and plant-derived natural products have been shown to possess membrane permeabilizing or efflux pump inhibitory activity, with the combination of azithromycin, ciprofloxacin, imipenem, etc. (**Aron and Opperman, 2016**; **Lin et al., 2015**; **Su and Wang, 2018**).

Artesunate (AS) is a semi-synthetic derivative of antimalarial compound artemisinin that is extracted from the traditional Chinese herb *Artemisia annua*. Beyond remarkable antimalarial action, AS and other artemisinin derivatives, for example, dihydroartemisinin, have been proven to restore the antibacterial effect of COL against *Escherichia coli*, while they themselves did not exhibit intrinsic antimicrobial activity against clinical *E. coli* isolates as well as ATCC 25922 (**Wei et al., 2020**; **Zhou et al., 2022**). In addition, AS has also been proven to enhance the effectiveness of various β-lactam and fluoroquinolones antibiotics against MDR *E. coli* via inhibiting the efflux pump AcrAB-TolC (**Pan et al., 2020**; **Wei et al., 2020**). Nonetheless, the synergistic effect between AS and COL was only observed in a limited number of strains, with a modest reversal effect. Therefore, few studies have been undertaken to evaluate its underlying mechanism. Under this circumstance, it is meaningful to

**Table 1.** The antibacterial activities of COL, AS, and EDTA against the tested strains after single and double combinations.

| | | MICs (mg/L) | | | | | | | | | | |
| | | Alone | | | COL + AS | | | | COL + EDTA | | | |
| Strains | *mcr-1* | COL | AS | EDTA | 1/4 AS | 1/8 AS | 1/16 AS | Fold change | 1/4 EDTA | 1/8 EDTA | 1/16 EDTA | Fold change |
|---|---|---|---|---|---|---|---|---|---|---|---|---|
| JS | − | 0.25 | 1250 | 125 | 0.0625 | 0.125 | 0.125 | 2–4 | 0.25 | 0.25 | 0.25 | 0 |
| S34 | − | 0.125 | 1250 | 125 | 0.008 | 0.015 | 0.125 | 0–16 | 0.015 | 0.03 | 0.0625 | 2–8 |
| S16 | − | 4 | 1250 | 125 | 0.25 | 0.25 | 4 | 0–16 | 2 | 2 | 4 | 0–2 |
| S20 | − | 2 | 1250 | 125 | 0.015 | 0.015 | 2 | 0–133 | 0.25 | 2 | 2 | 0–8 |
| S13 | + | 2 | 1250 | 125 | 0.25 | 0.25 | 2 | 0–8 | 1 | 2 | 2 | 0–2 |
| S30 | + | 4 | 1250 | 125 | 0.25 | 0.25 | 2 | 2–16 | 2 | 2 | 2 | 2 |
| E16 | + | 2 | 1250 | 500 | 0.0625 | 0.25 | 2 | 0–32 | 0.5 | 0.5 | 1 | 2–4 |
| M15 | − | >64 | 1250 | >1000 | >64 | >64 | >64 | 0 | >64 | >64 | >64 | 0 |
| P01 | − | >64 | 1250 | >1000 | >64 | >64 | >64 | 0 | >64 | >64 | >64 | 0 |

AS, artesunate; COL, colistin; MIC, minimum inhibitory concentration.

**Table 2.** The antibacterial activities of COL against the tested strains after single and triple combinations.

| Strains | mcr-1 | COL alone | COL + 1/4 AS + EDTA | | | | COL + 1/8 AS + EDTA | | | | Fold change |
| | | | 1/4 EDTA | 1/8 EDTA | 1/16 EDTA | 1/32 EDTA | 1/4 EDTA | 1/8 EDTA | 1/16 EDTA | 1/23 EDTA | |
| JS | − | 0.25 | 0.015 | 0.125 | 0.125 | 0.125 | 0.03 | 0.25 | 0.25 | 0.25 | 0–17 |
| S34 | − | 0.125 | 0.008 | 0.015 | 0.015 | 0.015 | 0.02 | 0.0625 | 0.125 | 0.125 | 0–16 |
| S16 | − | 4 | 0.015 | 0.03 | 0.0625 | 0.0625 | 0.0625 | 0.125 | 0.125 | 0.5 | 8–267 |
| S20 | − | 2 | 0.0625 | 0.125 | 0.125 | 0.125 | 0.25 | 0.5 | 0.5 | 0.5 | 4–32 |
| S13 | + | 2 | 0.015 | 0.125 | 0.25 | 0.25 | 0.0625 | 0.25 | 0.5 | 0.5 | 4–133 |
| S30 | + | 4 | 0.015 | 0.0625 | 0.0625 | 0.125 | 0.125 | 0.125 | 0.25 | 0.5 | 8–266.6 |
| E16 | + | 2 | 0.00003 | 0.002 | 0.25 | 0.5 | 0.004 | 0.015 | 0.25 | 1 | 2–66,667 |
| M15 | − | >64 | 32 | 32 | >64 | >64 | 32 | 32 | >64 | >64 | 0–2 |
| P01 | − | >64 | >64 | >64 | >64 | >64 | >64 | >64 | >64 | >64 | 0 |

AS, artesunate; COL, colistin; .

explore new drug combinations between AS and COL against MDR bacteria. Encouragingly, in this study, we confirmed the prominent synergistic effects of AS and EDTA to restore the antimicrobial activity of COL and its possible molecular mechanism.

## Results

### Artesunate and EDTA could enhance the effects of colistin against *Salmonella* strains

The antimicrobial activities of AS, EDTA, or COL alone were initially investigated in the COL-sensitive strains of *Salmonella* (JS, S34), COL-resistant clinical strains of *Salmonella* (S16, S20, S13, and S30), *E. coli* (E16), and intrinsically COL-resistant species (*Morganella morganii* strain M15, *Proteus mirabilis* strain P01). Results showed that AS or EDTA alone had no direct antibacterial activity against these strains, with the minimum inhibitory concentrations (MICs) 1250 or >125 mg/L (*Table 1*). Except for the two intrinsically COL-resistant strains M15 and P01, there was only a slight decrease in COL MICs for other strains (fold changes ranging from 0 to 133), when the subinhibitory concentrations (1/4, 1/8, 1/16 MIC) of AS or EDTA were combined with COL (namely AC or EC) (*Table 1*). Whereas we found a marked decrease in COL MICs (up to 60,000-fold) after three drug combinations (namely AEC) (*Table 2*). These results indicated that when used simultaneously with COL, AS, and EDTA exerted antibacterial enhancement activity for *Salmonella* and *E. coli*, but not the intrinsically COL-resistant species. Thus, AS and EDTA could be considered adjuvants to reverse COL resistance in *Salmonella*.

To verify the antibacterial enhancement activity of AS and EDTA, the growth curves of *Salmonella* JS, S16 (*mcr-1*-), and S30 (*mcr-1*+) for the combined treatments were generated within 24 hr (*Figure 1*). Generally, the higher concentration of COL (2 mg/L), either alone or in combinations, was found to be more effective against these strains than that of the lower concentration (0.1 mg/L). Compared with the control groups, when these bacteria were grown in the presence of COL alone (0.1 mg/L) or different drug combinations, the antimicrobial activity was not significant after incubation for 24 hr (*Figure 1a–c*). By contrast, higher concentration of COL (2 mg/L), alone or in combinations, showed better antibacterial activity, and a gradual increase in antibacterial activity was observed: AEC > AC > EC > C (*Figure 1d–f*). It is worth noting that the effect of different combinations against *mcr-1*+ strain S30 was weaker than that of *mcr-1*- S16 and standard sensitive strain JS.

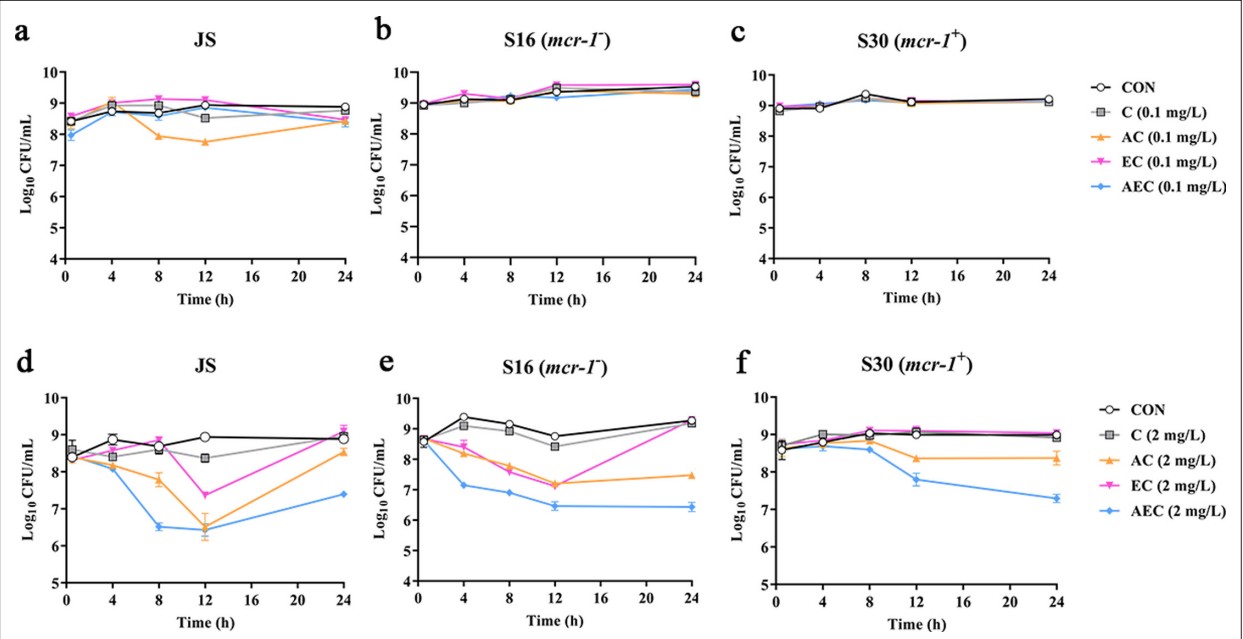

**Figure 1.** Time-kill curves of *Salmonella* strains JS, S16, and S30 with colistin (COL) alone and in combinations. (**a–f**) Samples were treated with different concentrations of COL (0.1 or 2 mg/L), alone or in drug combinations, for 12 hr. When used in combination, 1/8 minimum inhibitory concentration (MIC) of artesunate (AS) or EDTA was added to a final concentration of 156.3 or 15.6 mg/L, respectively. Counts of CFU/mL were performed on all cultures at each time point, and data are mean ± SD from representative of three independent experiments. CON indicates the negative control group.

## Artesunate and EDTA enhanced the membrane-damaging effect of colistin on *Salmonella*

In order to gain insight into the membrane-damaging bactericidal mechanism of COL alone or combinations, the damages to the bacterial outer and inner membrane (OM and IM) were severally monitored by measuring the fluorescent intensity of *Salmonella* strains S16 and S30 mixed with 1-N-phenylnaphthylamine (NPN) and propidium iodide (PI). Overall, the fluorescence signals of NPN and PI increased progressively with increases in the concentration of COL. After the treatment of AEC, S16, and S30 both showed the strongest fluorescence signals of NPN compared to those of other groups (*Figure 2a and b*). Nevertheless, AS treatment group also exerted a significant increase in fluorescent signal, although it is not as strong as that of the AEC-treated group. Meanwhile, there were rapid and significant increases in the fluorescence of PI when AEC or EC as added to bacterial cultures, and the two regimens played dominant roles in low or high concentrations of COL, respectively (*Figure 2c and d*). Therefore, these results indicated that the bacterial cell surfaces were severely damaged after the AEC treatment via the rapid perturbations of OM and IM. Subsequently, the morphological changes of *Salmonella* strain S16 treated with different regimens were further investigated using scanning electron microscope (SEM) analysis to confirm the above membrane-damaging effect. As shown in *Figure 3*, SEM micrographs of control and solvent-treated groups revealed that cells were short and rod-shaped, with rounded ends and intact cell membranes. As expected, after exposure to COL alone and different regimens, noticeable damage to the OM was observed, especially in the AC and AEC-treated groups. Exposure to AC and AEC leads to cell damages characterized by folds, crevices, and depressions, which further suggested that AS and EDTA combined with COL resulted in remarkable bacterial membrane injury for antibacterial activity. Collectively, these data confirmed the membrane-damaging effects of AS, EDTA, and COL combination against *Salmonella*.

## AEC combination could collapse the Δψ component of proton motive force (PMF) in *Salmonella*

In bacteria, the PMF, alternatively known as electrochemical proton gradient, results from the extrusion of protons by the electron transport chain and is made up of the sum of two parameters: an electric potential ($\Delta \phi$) and an osmotic component (ΔpH) (*Farha et al., 2013*). It has been reported to drive

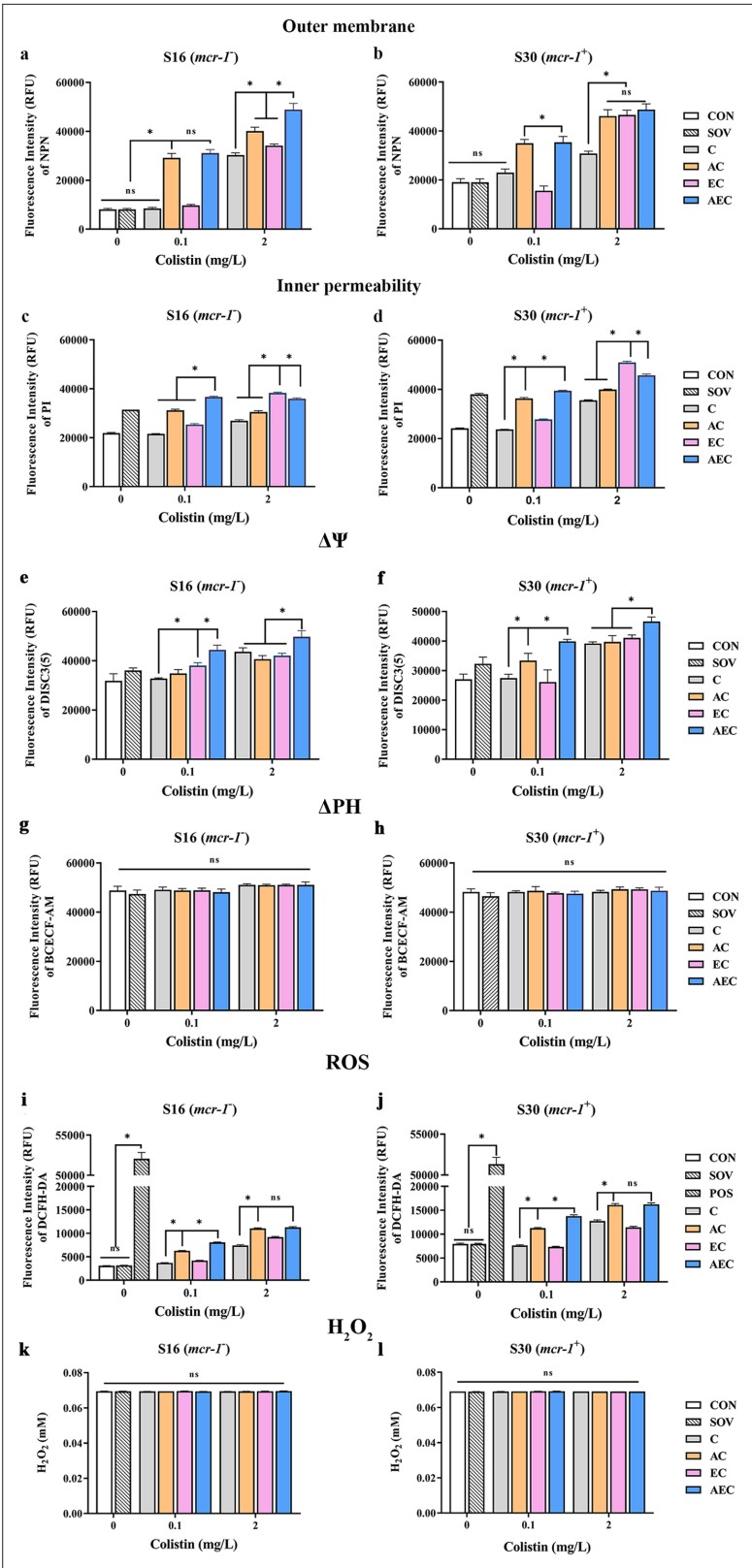

**Figure 2.** Artesunate (AS), EDTA, and colistin (COL) affected membrane integrity, proton motive force (PMF), reactive oxygen species (ROS), and $H_2O_2$ levels in S16 and S30 strains. Different concentrations of COL (0.1 or 2 mg/L) were used alone or in combination with AS and EDTA. When used in combination, 1/8 minimum inhibitory concentration (MIC) of AS or EDTA was added to a final concentration of 156.3 or 15.6 mg/L, respectively. a-d

*Figure 2 continued on next page*

*Figure 2 continued*

AS, EDTA, and COL affected membrane integrity as measured by fluorescence probes 1-N-phenylnaphthylamine (NPN) and propidium iodide (PI). Error bars indicate SDs for three replicas (*p<0.001, ns not significant). CON indicates the negative control group, and SOV indicates the solvent-exposed group. (**e–h**) Disruption of PMF is shown by measuring the dissipation of electric potential ($\Delta\psi$) (**a, b**) and osmotic component (ΔpH) (**c, d**). Error bars indicate SDs for three replicas (*p<0.001, ns not significant). CON indicates the negative control group, and SOV indicates the solvent-exposed group. (**i–l**) Intracellular accumulation of ROS (**i, j**) and $H_2O_2$ (**k, l**) in S16 and S30 strains after 1 hr treatment. Data are shown as the mean of triplicates ± SD (*p<0.001, ns not significant). CON indicates the negative control group, SOV indicates the solvent-exposed group, and POS indicates the positive control group that were treated with Rosup from the Total ROS Detection Kit.

The online version of this article includes the following figure supplement(s) for figure 2:

**Figure supplement 1.** Intracellular accumulation of reactive oxygen species (ROS) in S16 and S30 strains after 6 hr treatment.

vital cellular processes in bacteria, including ATP synthesis, antibiotic transport, and cell division (*Le et al., 2021*). Therefore, PMF dissipation was considered a promising strategy for combating microbial pathogens. In this work, we explored whether the antibacterial synergism activities of different combinations were accompanied by the dissipation of PMF in cells. We uncovered that compared to that of other groups, AEC treatment caused a rapid collapse of $\Delta\psi$ component, as shown by the increase in fluorescence values of $DISC_3(5)$ (*Figure 2e and f*). But none of these combinations observed significant dissipation of ΔPH compared to that of the control group (*Figure 2g and h*). The above results indicated that AEC treatment was able to dissipate selectively the $\Delta\psi$ component of PMF.

## Other reactive oxygen species (ROS), not $H_2O_2$, contributed to the AS and EDTA-mediated efficacy enhancement of COL

ROS, including superoxide ($O^{2-}$), hydrogen peroxide ($H_2O_2$), and hydroxyl radical (·OH), are commonly generated during the electron transfer process, which has been considered to be associated with the lethal action of diverse antimicrobials. Subsequently, we investigated whether the addition of AS and EDTA could facilitate the intracellular ROS generation and stimulate the ROS-mediated killing. As shown in *Figure 2i and j*, compared to the control group, total ROS increases in S16 and S30 strains were observed in AC and AEC groups either after 0.1 or 2 mg/L COL was added, but for COL alone and EC groups, the increase only occurred after the addition of a relatively high COL concentration (2 mg/L). In addition, the ROS accumulation level was further increased in the AEC group when extending the incubation time period to 6 hr (*Figure 2—figure supplement 1*). Moreover, we found that $H_2O_2$ did not contribute to the increase in total ROS as there was no significant difference in the intracellular level of $H_2O_2$ among all the groups (*Figure 2k and l*). Collectively, these observations supported a role for ROS in AS and EDTA-mediated efficacy enhancement of COL.

## The transcriptome data exhibited more robust changes than that of metabolome among different comparison groups

A total of 6944 differentially expressed genes (DEGs) were performed KEGG pathway enrichment analysis, of which 1832 and 5112 transcripts were included in S16 and S30 strains, respectively (*Figure 4—figure supplement 1*). Several canonical pathways, including two-component system (TCS), flagellar assembly, and ABC transporters pathways, indicating similar directional changes in both strains were selected for further analysis, as shown in *Figure 4* and *Figure 4—figure supplement 2*. Since AEC incubation has displayed excellent antibacterial effects, we expected to screen the significantly differentially expressed genes (SDEGs) with similar variations in AEC vs. C, AEC vs. AC, and AEC vs. EC groups, and the SDEGs involved in TCS (*pagC, cheA, ompF*, etc.), flagellar assembly (*flgK, flgL, fliD*, etc.), and ABC transporters (*oppuBB, osmX, gltI, dppA*, etc.) were selected (mostly downregulated) and summarized in *Figure 5*.

Unlike transcriptome changes, the metabolite alterations were much less abundant among AEC vs. C, AEC vs. AC, and AEC vs. EC groups, either in S16 or S30 strain. Unluckily, we demonstrated that there was a low correlation between the metabolome and transcriptome data. According to the enrichment analysis, arachidonic acid metabolism (downregulated), degradation of aromatic

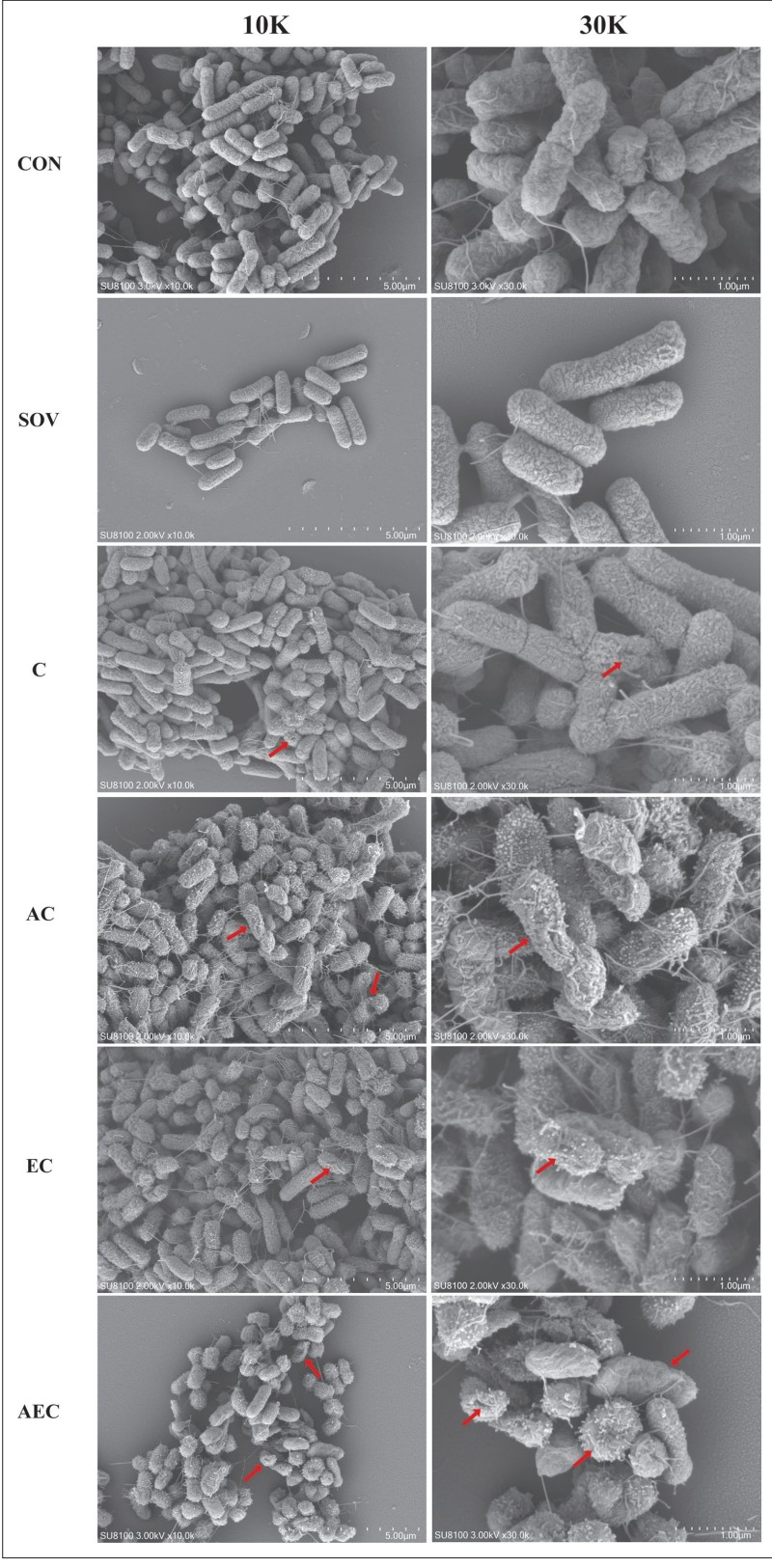

**Figure 3.** Morphological changes of S16 (*mcr-1*) strain. The images were obtained after the treatment with colistin (COL) (2 mg/L) alone or in combination with 1/8 minimum inhibitory concentration (MIC) of artesunate (AS) (156.3 mg/L) or EDTA (15.6 mg/L). CON indicates the negative control group, and SOV indicates the solvent-exposed group. Red arrows indicate the cell damages characterized by folds, crevices, or depressions.

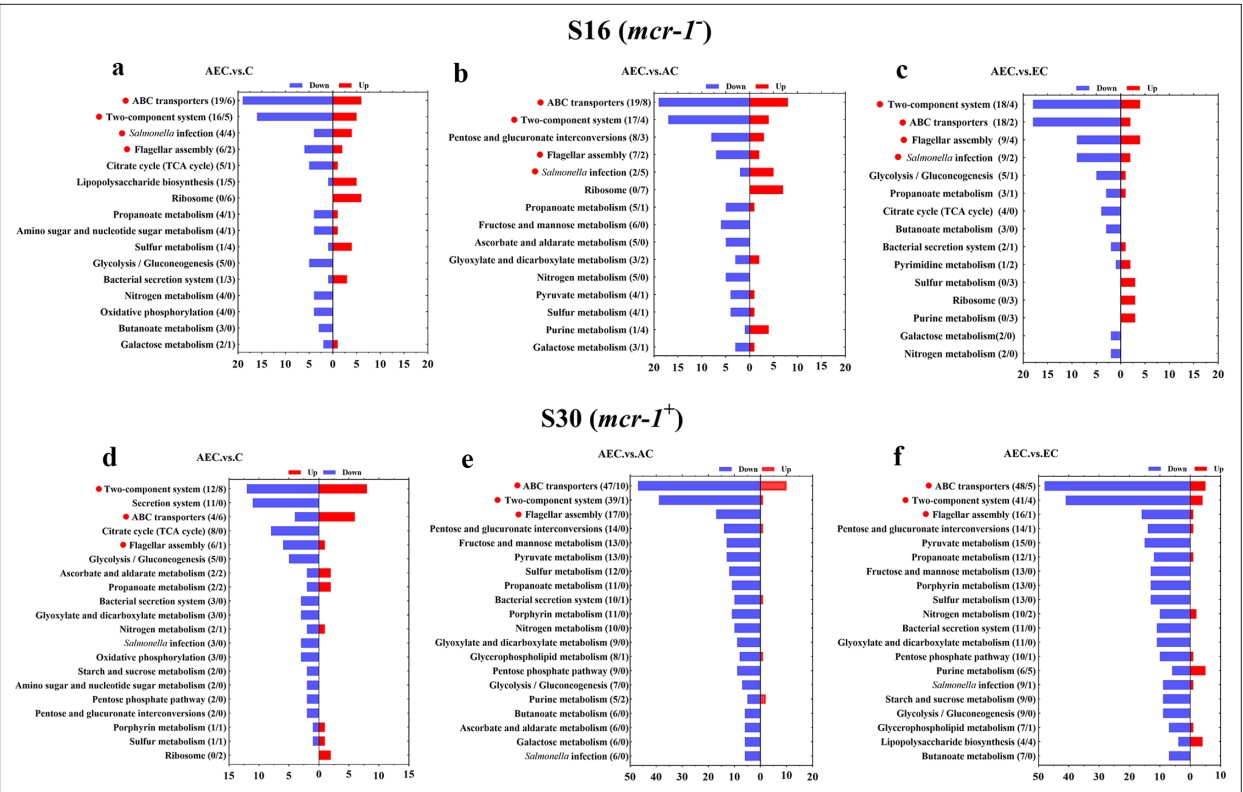

**Figure 4.** Kyoto Encyclopedia of Genes and Genomes (KEGG) pathway analysis of significantly differentially expressed genes (SDEGs) in S16 (**a–c**) and S30 (**d–f**) strains within the AEC. vs. C, AEC.vs. AC, and AEC.vs. EC groups. Samples were harvested after the treatment of COL (2 mg/L) alone or in combination with 1/8 MIC of AS (156.3 mg/L) or EDTA (15.6 mg/L) for 6 hr. Pathway name and number of downregulated (blue) and upregulated (red) genes in each pathway are indicated in parentheses on the left (down/up). Highlighted with red circles are the pathways where SDEGs are mainly enriched and appeared simultaneously in different comparison groups.

The online version of this article includes the following figure supplement(s) for figure 4:

**Figure supplement 1.** The number of differentially expressed genes (DEGs) is identified in S16 and S30 strains among different comparison groups.

**Figure supplement 2.** Kyoto Encyclopedia of Genes and Genomes (KEGG) pathway analysis of significantly differentially expressed genes (SDEGs) in S16 (**a, b**) and S30 (**c, d**) strains within the AC. vs. C, and EC.vs. C groups.

compounds (upregulated), taurine and hypotaurine metabolism (upregulated) were the most prominent pathways showing differences primarily in the AEC vs. C group (*Figure 6*).

## AS + EDTA + COL combination therapy is a promising therapeutic against *Salmonella* infection in vivo

The excellent bactericidal synergism against *Salmonella* in vitro of AEC combination further prompted us to confirm the effect in vivo for *Salmonella* S30 (*mcr-1*$^+$)-infected mouse models. Consistent with the synergistic bactericidal activity of AS, EDTA, and COL, the combination of AC (7.13 log$_{10}$CFU/g liver, 6.91 log$_{10}$CFU/g spleen), EC (7.33 log$_{10}$CFU/g liver, 6.88 log$_{10}$CFU/g spleen), and AEC (6.51 log$_{10}$CFU/g liver, 6.52 log$_{10}$CFU/g spleen) outperformed single-drug treatments of COL (7.44 log$_{10}$CFU/g liver, 7.05 log$_{10}$CFU/g spleen) and AS (7.54 log$_{10}$CFU/g liver, 7.02 log$_{10}$CFU/g spleen). In particular, in the AEC combination-treated samples, there were far fewer bacteria burden in the spleen and liver compared to other groups (*Figure 7*). This disparity in vivo further illustrates the synergy of AS and EDTA in combination with COL.

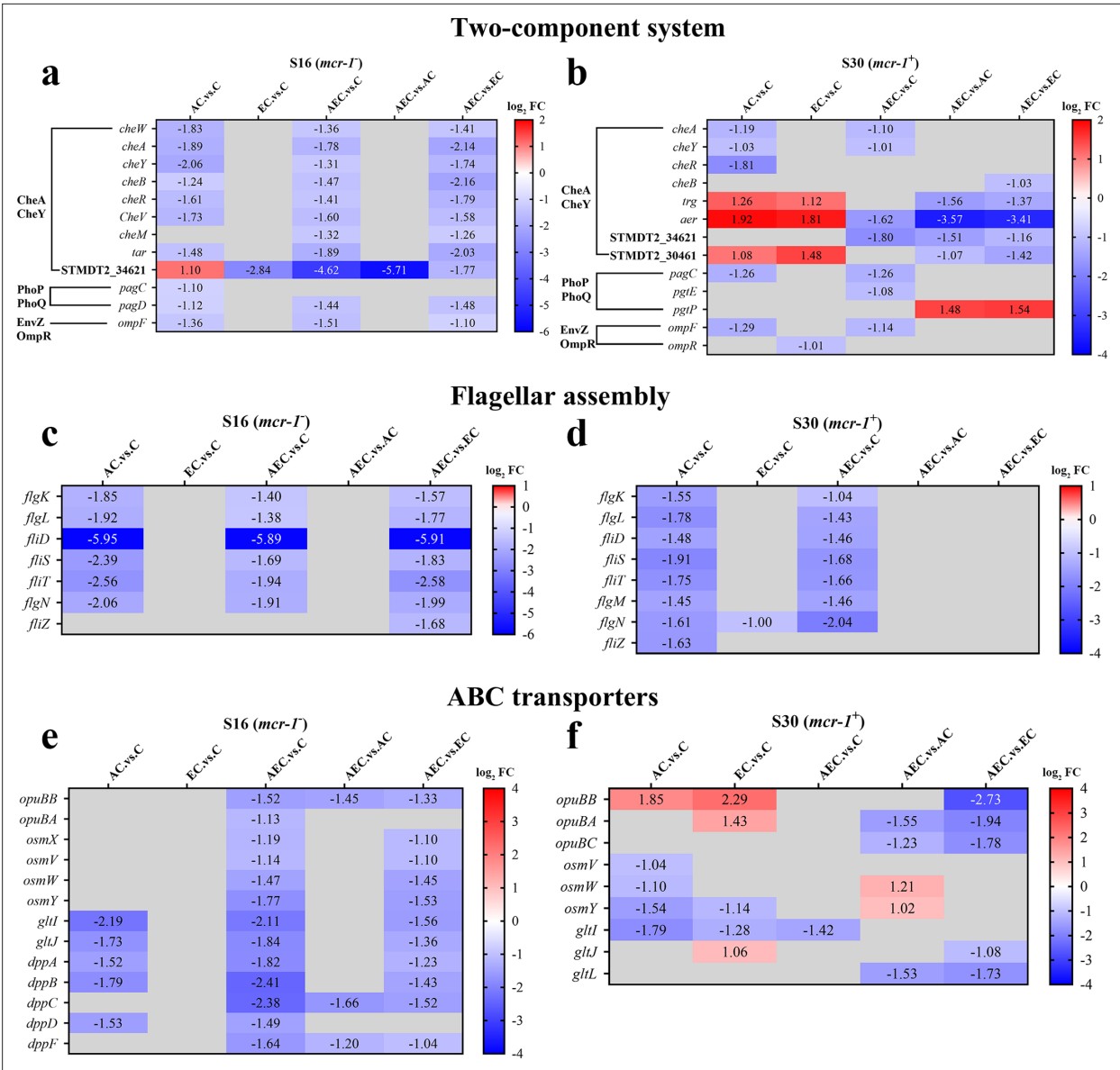

**Figure 5.** The significantly differentially expressed genes (SDEGs) detected in two-component system (**a, b**), flagellar assembly (**c, d**), and ABC transporters (**e, f**) pathways among different comparison groups, within S16 and S30 strains. Samples were harvested after the treatment of colistin (COL) (2 mg/L) alone or in combination with 1/8 minimum inhibitory concentration (MIC) of artesunate (AS) (156.3 mg/L) or EDTA (15.6 mg/L) for 6 hr. Labels in each square indicate the $\log_2$ (fold change) of corresponding genes. Squares without label and gray background indicate the data are not credible (p>0.05, |$\log_2$Fold Change|<1.0). Background colors indicate the expression levels of the respective genes, red = upregulated, blue = downregulated. $\log_2$FC: $\log_2$Fold Change.

The online version of this article includes the following figure supplement(s) for figure 5:

**Figure supplement 1.** Artesunate (AS), EDTA, and colistin (COL) inhibited the swimming motility of S16 (**a**) and S30 (**b**) strains.

**Figure supplement 2.** The significantly differentially expressed genes (SDEGs) detected in *Salmonella* infection and ribosome pathways (**a**); expression level of *mcr-1* (**b**); efflux pump activity in S16 or S30 strain (**c, d**); putative pattern of interaction between artesunate (AS) and MCR-1 protein (**e**).

## Discussion

### Membrane damaging played a crucial role in the synergistic antimicrobial effects of AS + EDTA + COL combination

COL is an increasingly important antibiotic against serious infections caused by G⁻ bacteria. It damages both the OM and IM layers of the cell surface by targeting LPS, displacing cations that form bridges

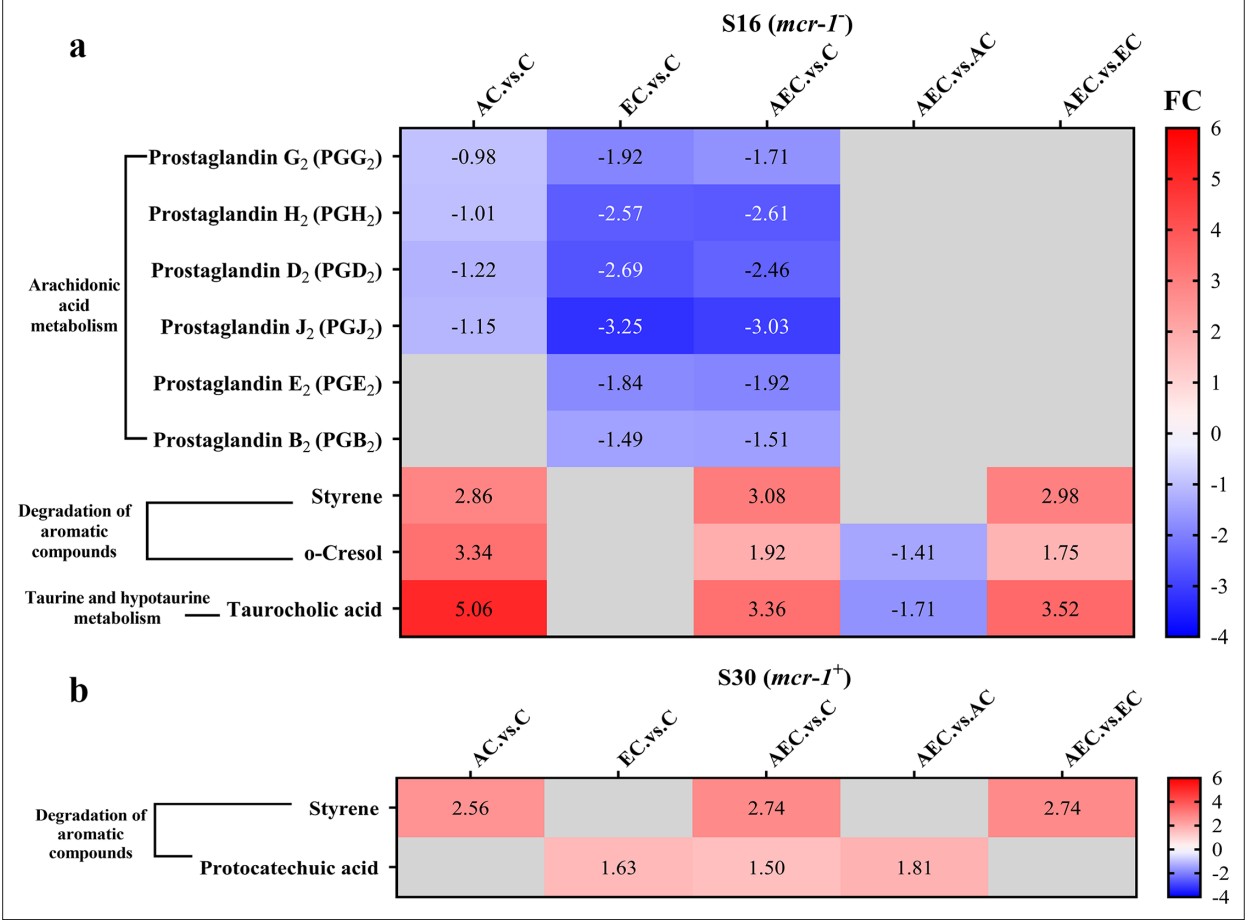

**Figure 6.** The significant differential metabolites (SDMs) detected in arachidonic acid metabolism, degradation of aromatic compounds, taurine and hypotaurine metabolism pathways among different comparison groups, within S16 (**a**) and S30 (**b**) strains. Samples were harvested after the treatment of colistin (COL) (2 mg/L) alone or in combination with 1/8 minimum inhibitory concentration (MIC) of artesunate (AS) (156.3 mg/L) or EDTA (15.6 mg/L) for 6 hr. Labels in each square indicate the fold changes of corresponding metabolites. Squares without label and gray background indicate the data are not credible (VIP < 1.0, 0.833 > Fold Change < 1.2 or Fold Change ≤ 0.833, p≥0.05). Background colors indicate the fold changes of the respective metabolites, red = increased, blue = decreased.

between LPS molecules, and thereby leading to disruption of the cell envelope and bacterial lysis (*Sabnis et al., 2021*). Our data showed that the AEC combination could permit ingress of NPN fluorophore into the OM, as well as the membrane impermeant dye PI into the IM, which fluoresces upon contact with DNA in the bacterial cytoplasm (*Figure 2a–d*). Thus, it indicated AEC combination has punched holes in both the OM and IM of whole bacterial cells. The considerably deformed cell membranes observed by SEM further supported the above results that the membrane integrity and permeability were damaged after AEC incubation. Considering that EDTA is used as a complexing agent, we tried to explore if it could assist COL in destroying the bacterial membrane and reduce their survival by chelating cations that stabilize LPS and the outer membrane. Nevertheless, we found minimal changes in MICs of different combinations to S16 and S30 strains, after different cations ($Na^+$, $K^+$, $Ca^{2+}$, $Mg^{2+}$, $Mn^{2+}$, $Zn^{2+}$) were supplemented (*Table 3*). In contrast, LPS treatment resulted in dose-dependent changes in MICs of different combinations to S16 and S30 strains (*Table 4*), which indicated that LPS on cell membrane was a crucial target for AEC to injure cell membrane and exert prominent antibacterial effects.

While it is commonly believed that COL acts against G⁻ bacteria by cell membrane lysis, we hypothesized that alterations in bacterial membrane lipid composition may also be a possible Achilles' heel to increase the efficacy of COL after the AEC incubation in this article. The metabolomic results of AEC vs. C group in this study showed a significant decrease in prostaglandins (PGs) in arachidonic acid metabolism pathway (*Figure 6*). Arachidonic acid is a highly abundant long-chain polyunsaturated fatty acid

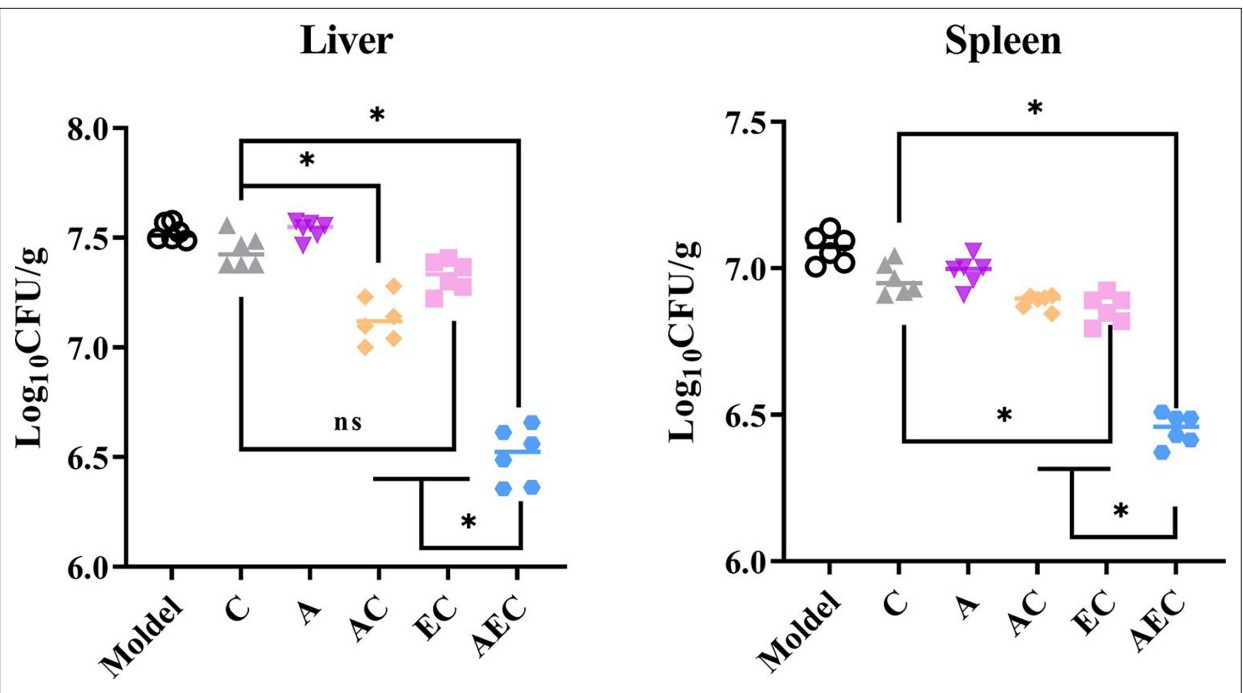

**Figure 7.** Artesunate (AS) and EDTA potentiate colistin activity against *Salmonella* S30 (*mcr-1*⁺) in vivo. Kunming mice (n = 6 per group) were intraperitoneally given a nonlethal dose of *Salmonella* S30 (1.31 × 10⁵ CFU), then treated with PBS, colistin (COL) (10 mg/kg), AS (15 mg/kg), AS (15 mg/kg) + COL (10 mg/kg), EDTA (50 mg/kg) + COL (10 mg/kg), and AS (15 mg/kg) + EDTA (50 mg/kg) + COL (10 mg/kg) by intraperitoneal injection. Bacterial loads were determined in the spleen and liver and bacterial counts were computed and presented as the mean ± SD log₁₀ CFU/mL. The p-values were determined by one-way ANOVA (*p<0.001, ns, not significant).

**Table 3.** Minimum inhibitory concentration (MIC) values of colistin (COL) against S16 and S30 strains after the addition of exogenous cations.

| Strains | Cations (100 mg/L) | MICs (mg/L) | | | |
|---|---|---|---|---|---|
| | | COL | AC | EC | AEC |
| S16 (*mcr-1⁻*) | Control | 4 | 0.25 | 2 | 0.125 |
| | Na⁺ | 4 | 0.25 | 2 | 0.125 |
| | K⁺ | 4 | 0.25 | 2 | 0.125 |
| | Ca²⁺ | 4 | 0.25 | 2 | 0.125 |
| | Mg²⁺ | 8 | 0.5 | 4 | 0.25 |
| | Mn²⁺ | 4 | 0.25 | 2 | 0.125 |
| | Zn²⁺ | 4 | 0.25 | 2 | 0.125 |
| S30 (*mcr-1⁺*) | Control | 4 | 0.25 | 2 | 0.125 |
| | Na⁺ | 4 | 0.25 | 2 | 0.125 |
| | K⁺ | 4 | 0.25 | 2 | 0.125 |
| | Ca²⁺ | 4 | 0.25 | 2 | 0.125 |
| | Mg²⁺ | 8 | 0.5 | 4 | 0.25 |
| | Mn²⁺ | 4 | 0.25 | 2 | 0.125 |
| | Zn²⁺ | 4 | 0.25 | 2 | 0.125 |

**Table 4.** Minimum inhibitory concentration (MIC) values of colistin (COL) against S16 and S30 strains after the addition of exogenous LPS.

| Strains | Drug | MICs (mg/L) | | | |
| --- | --- | --- | --- | --- | --- |
| | | +0 LPS | +4 LPS | +32 LPS | +512 LPS |
| S16 (*mcr-1⁻*) | COL | 4 | 4 | 8 | 64 |
| | AC | 0.25 | 0.5 | 2 | 32 |
| | EC | 2 | 2 | 2 | 32 |
| | AEC | 0.125 | 0.5 | 2 | 32 |
| S30 (*mcr-1⁺*) | COL | 4 | 4 | 16 | 64 |
| | AC | 0.25 | 0.5 | 4 | 64 |
| | EC | 2 | 4 | 16 | 64 |
| | AEC | 0.125 | 0.5 | 4 | 64 |

in vertebrates, which have been proposed to have antibacterial roles. Exogenous arachidonic acid has been reported to be readily incorporate into the synthesis pathways of membrane phospholipids, and exert detrimental effects on membrane integrity by perturbing membrane ordering, altering membrane composition, and increasing fluidity (*Eijkelkamp et al., 2018*; *MacDermott-Opeskin et al., 2022*). PGs are lipid compounds derived from arachidonic acid, which has been demonstrated to enhance biofilm development and fungal load in the murine vaginae of *Candida albicans* (*Ells et al., 2011*). Therefore, we speculated that the downregulated lipid compounds PGs may lead to perturbation of membrane phospholipids in cell membranes and reduce microbial viability.

Accumulation of toxic compounds may also be held responsible for the membrane damage. We noted that there was a significant accumulation of styrene in both S16 and S30 strains (*Figure 6*). Styrene is naturally present as a minor metabolite that can be synthesized at low levels by several microorganisms, like *Pencillium camemberti* and members of the *Styracaceae* family (*Yeh et al., 2022*). However, styrene itself is toxic to most cell types, and its hydrophobic molecules could readily partition into bacterial membrane, resulting in membrane disruption and cell death (*Lian et al., 2016*). Additionally, we found that the taurocholic acid and protocatechuic acid were upregulated respectively in S16 and S30 strains (*Figure 6*). Taurocholic acid is usually a major component of the selective culture medium, for example, MacConkey agar, for G⁻ bacteria, which has also been confirmed as a secondary metabolite of marine isolates and the soil bacterium *Streptococcus faecium*. Sannasiddappa et al. proved that taurocholic acid was able to inhibit the growth of *Staphylococcus aureus* by increasing membrane permeability and disruption of the PMF (*Sannasiddappa et al., 2017*). Protocatechuic acid has been demonstrated to exert antimicrobial effects by disrupting the cell membranes and preventing bacterial adhesion and biofilm formation (*Bernal-Mercado et al., 2018*; *Stojković et al., 2013*). Consequently, these toxic compounds were anticipated to accelerate the destruction of cell membrane and thus enhance the antibacterial activity.

### The CheA of chemosensory system and virulence-related protein SpvD were critical for the bacteriostatic synergistic effect of AEC combination

Flagellar motility is intimately connected to chemotaxis, biofilm formation, colonization, and virulence of many bacterial pathogens. It is generally regulated by a chemotactic signaling system, which enables their movement toward favorable conditions and invade their hosts (*Bolton, 2015*). The chemotaxis proteins CheA, CheW, CheY, and methyl-accepting chemotaxis proteins (MCPs) have already been identified as core components and present in all chemotaxis systems (*Minamino et al., 2022*). The *Salmonella* flagellum is composed of about 30 different proteins, such as FliD (the filament cap), FlgK and FlgL (the hook-filament junction), etc. (*Minamino et al., 2022*). We found an extensive downregulation of chemotaxis and flagellar assembly-related genes in S16 and S30 strains after AEC incubation (*Figure 5a–d*). Meanwhile, the swimming motility of the strains was decreased after AEC

**Table 5.** Minimum inhibitory concentration (MIC) values of colistin against S16 strain after the overexpression of different genes.

| Genes | L-Ara | MICs (mg/L) | | |
|---|---|---|---|---|
| | | AC | EC | AEC |
| | — | 0.25 | 0.5 | 0.125 |
| cheA | + | 0.25 | 2 | 4 |
| | — | 0.25 | 2 | 0.125 |
| cheY | + | 0.5 | 2 | 0.125 |
| | — | 0.25 | 1 | 0.125 |
| STMDT2-34621 | + | 0.5 | 1 | 0.25 |
| | — | 0.25 | 1 | 0.125 |
| aer | + | 0.25 | 1 | 0.125 |
| | — | 0.25 | 1 | 0.25 |
| fliD | + | 0.5 | 1 | 0.5 |
| | — | 0.25 | 1 | 0.125 |
| fliT | + | 0.25 | 1 | 0.125 |
| | — | 0.25 | 1 | 0.125 |
| opuBB | + | 0.25 | 1 | 0.125 |
| | — | 0.25 | 1 | 0.125 |
| gltI | + | 0.25 | 1 | 0.125 |
| | — | 0.25 | 1 | 0.125 |
| dppB | + | 0.25 | 1 | 0.125 |
| | — | 0.25 | 1 | 0.125 |
| dppC | + | 0.25 | 1 | 0.125 |
| | — | 0.25 | 1 | 0.125 |
| spvD | + | 1 | 1 | 0.5 |

incubation (**Figure 5—figure supplement 1**). The overexpression of cheA in S16 strain, but not cheY, STMDT2-34621, aer, fliD, fliT, could lead to noticeable increases of MICs by 4–32 fold, after EC or AEC incubation (**Table 5**). These results suggested that the downregulation of the central component of chemosensory system CheA may affect both chemotactic motility and general structure of flagellum, thus attenuating Salmonella survival after AEC treatment.

In addition to the above findings, we also noted that there was a relatively large number of SDEGs, most of them were downregulated, enriched in the ABC transports pathway in S16 and S30 strains after

AEC incubation (*Figure 5e and f*). ABC transporters are a class of transmembrane transporters that mediate the uptake of micronutrients, including saccharides, amino acids, and metal ions, and have also been shown to protect bacteria from hazardous compounds (*Nguyen and Götz, 2016*; *Wang et al., 2023*). The *opuBB* and *opuBA* encode components of the ABC-type proline/glycine betaine transport system, and their upregulation was proved to promote accumulation of proline that acted as an osmoprotectant (*Dupre et al., 2019*). Similarly, the OsmU osmoprotectant systems, consisting of OsmV, OsmW, OsmY, and OsmX, were identified to enable bacterial survival at high osmolarity through the accumulation of glycine betaine (*Frossard et al., 2012*). The *gltI* gene encodes glutamate/aspartate transport protein, and its deletion in *E. coli* was shown to result in attenuated survival under antibiotics, acid, and hyperosmotic stressors (*Niu et al., 2023*). Furthermore, the dipeptide permease operon (*dpp*), especially *dppA*, has been reported as an essential enzyme for the survival of *Mycobacteria tuberculosis* under nutrient starvation conditions, and was associated with reduced bacterial burden in chronically infected mice in knockout studies (*Fernando et al., 2022*). Nevertheless, the overexpression of *opuBB*, *gltI*, *dppB*, and *dppC* genes caused no detectable changes in MICs for S16 strain after AEC incubation (*Table 5*). Thus, these observations suggest that overexpression of these SDEGs in ABC transporters was not sufficient enough for causing changes in COL susceptibility, and additional factors may be required for the excellent bactericidal activity of AEC.

Besides, we also noticed that the expression levels of *Salmonella* infection-related genes *sseJ*, *spvD,* and ribosomal protein genes *rpmJ*, *rpmE* were all strikingly changed in the S16 strain (*Figure 5—figure supplement 2a*). SseL and SpvD are effectors of *Salmonella* pathogenicity islands 1 and 2 (SPI1 and SPI2), which are required for full virulence during animal infections (*Coombes et al., 2007*; *Grabe et al., 2016*). The *rpmJ* gene codified a part of 50S ribosomal subunit, and its upregulation has been related to size increase and slow growth of *E. coli* cells under starvation conditions (*Peredo-Lovillo, et al., 2019*). The overexpression of *spvD* gene in S16 strain could attenuate the antibacterial activity of AC and AEC regimen with fourfold increase in MICs (*Table 5*), which indicated that the virulence-related protein SpvD contributed to the increase in COL susceptibility after AEC incubation.

## Artesunate can be considered a potential MCR-1 inhibitor that enhances the efficacy of colistin

During our research, several phenomena caught our attention such as the synergistic effect of AEC combination was irrelevant to whether the *mcr-1* gene exists or not, and the *mcr-1*⁻ strain S16 exhibited more robust changes than that of *mcr-1*⁺ strain S30 in different KEGG pathways. These results indicated that AS and EDTA were possible to exert synergistic effects by blocking the broad-spectrum resistance mechanisms (e.g., efflux pumps, membrane damage), and coupling with the drug-specific resistance mechanisms (e.g., MCR-1, β-lactamase). Previously, AS was capable of significantly enhancing the antibacterial activity of β-lactam antibiotics against *E. coli* via inhibition of the efflux pumps such as AcrB, NorA, NorB, and NorC (*Jiang et al., 2013*; *Li et al., 2011*). Molecular docking experiments showed that AS could dock into AcrB very well by forming five hydrogen bonds with Ser46, Gln89, and Gln176 (*Wu et al., 2013*). Whereas, in this article, the inhibitory actions of different drug combinations on efflux pump were only observed in *mcr-1*⁻ strain S16, regardless of whether AS was added (*Figure 5—figure supplement 2c*). Therefore, we hypothesized that AS may exert synergistic effects with COL against *mcr-1*⁺ S30 strain by targeting MCR-1 rather than efflux pump.

MCR-1 comprises two distinct domains, an N-terminal transmembrane domain and a soluble C-terminal α/β/α sandwich domain where the active site is located. The active site contains a concentration of metal-binding residues to accommodate between one and four zinc ions (*Wei et al., 2018*). Several crystal structures of the soluble domain have been well studied, and six residues GLU246, THR285, HIS395, ASP465, HIS466, and HIS478 were found to be conserved among pEtN transferases (*Ma et al., 2016*). Especially the THR285 residue, which was highly conserved and provided a distinct electronegative potential to attract and bind the substrate pEtN (*Son et al., 2019*). Mutations of these residues and stripping the metals by EDTA could re-establish polymyxin B antibacterial action (*Hinchliffe et al., 2017*; *Hu et al., 2016*; *Stojanoski et al., 2016*). In this article, we first analyzed the relative expression of *mcr-1* in S30 strain after the incubation of different drug combinations and found a striking downregulation of *mcr-1* gene whether after AC, EC incubation, or AEC incubation, compared to that of A and E treatment (*Figure 5—figure supplement 2b*). Additionally, we performed the molecular docking between AS and MCR-1 to predict if there were possible interactions and found that AS

**Table 6.** Minimum inhibitory concentration (MIC) values of colistin against S30 strain after the incubation of polypeptides.

| Strains | Drug | MICs (mg/L) | | |
| --- | --- | --- | --- | --- |
| | | Control | +$P_u$ | +$P_m$ |
| S30 (*mcr-1*[+]) | AC | 0.25 | 2 | 0.25 |
| | AEC | 0.125 | 1 | 1 |

$P_u$ and $P_m$ indicate the unmutated and mutated peptide at THR 283, SER 284, and TYR 287 sites, respectively.

could bind to the residues surrounding the reported key residues within MCR-1 of *E. coli*, forming seven hydrogen bonds with THR283, SER284, TYR287, PRO481, and ASN482 residues (*Figure 5—figure supplement 2e*). The interaction between AS and MCR-1 was further proved by competitive inhibitory assays that the binding of AS with MCR-1 was blocked after the addition of polypeptide $P_u$ (containing unmutated THR283, SER284, and TYR287 sites) and led to significant increases of MICs by eightfold after AC or AEC treatment (*Table 6*). Nevertheless, we also noticed that the blocking effect could not be removed by peptide $P_m$ (containing mutated THR283, SER284, and TYR287 sites) when EDTA exists. We supposed that EDTA may chelate zinc ions that are required for MCR-1 activity. Thus, we suggested that AS could be developed as an MCR-1 inhibitor, and coupled with the chelating agent EDTA may favor additionally magnifying its inhibitory effect.

## A mixed blessing: The excellent antibacterial activity and potential toxicity of AEC combination

In the preceding years, several studies have reported on the synergistic effects of COL with different candidates, such as antimicrobial agents, natural compounds, and synthetically prepared molecules (*Cui et al., 2024*; *Yi et al., 2022*). Nevertheless, the potential toxicity of the combination therapy will be the prime concern affecting their clinical application, and a similar concern has also been raised for the AEC combination in this study. Although in vitro studies have determined that with increasing doses of AS and EDTA the antibacterial synergistic activity was gradually enhanced, and meanwhile, may also result in more toxic side effects. Thus, in our study, the 1/8 MICs of AS and EDTA were selected to ensure excellent antibacterial activity whereas minimizing the potential toxicity. The toxic side effects of AEC combination may be most probably caused by COL, which is well known to present several adverse toxic effects, and the dose-dependent nephrotoxicity is the most reported (*Lu et al., 2016*; *Visentin et al., 2017*). Conversely, EDTA and AS may be low toxicity. When used in combination with COL or AB569, EDTA has been proven to synergistically overcome the COL-resistant *Klebsiella pneumoniae* and MDR *Acinetobacter baumannii* with the concentration of 12,000 mg/L and 73 mg/L, respectively (*Bari et al., 2023*; *Bogue et al., 2021*). These concentrations were reported to be nontoxic to primary adult human skin (dermal) fibroblasts and other normal body cells, which may also apply to the lower concentrations of EDTA (15.6 mg/L) used in AEC combination (*Shein et al., 2021*). Additionally, AS has been found to exert multiple pharmacological actions including antimalaria, antitumor, antiviral, and anti-inflammatory effects, which has displayed a relatively safe toxicity profile with the $LD_{50}$ values being 4223 mg/kg (*Cheong et al., 2020*). However, the potential toxic effects of AEC combination are still unknown and require further investigations.

In summary, our results established that the combination of COL with AS and EDTA was a promising candidate for combating infections caused by MCR-negative and -positive COL-resistant *Salmonella*. The membrane-damaging effect, accumulation of toxic compounds, and inhibition of MCR-1 were supposed to play synergistic roles in reversing COL resistance of *Salmonella* (*Figure 8*). The CheA of chemosensory system and virulence-related protein SpvD were critical for the bacteriostatic synergistic effects of AEC combination. Selectively targeting CheA, SpvD, or MCR using the natural compound AS could be further investigated as an attractive strategy for the treatment of *Salmonella* infection.

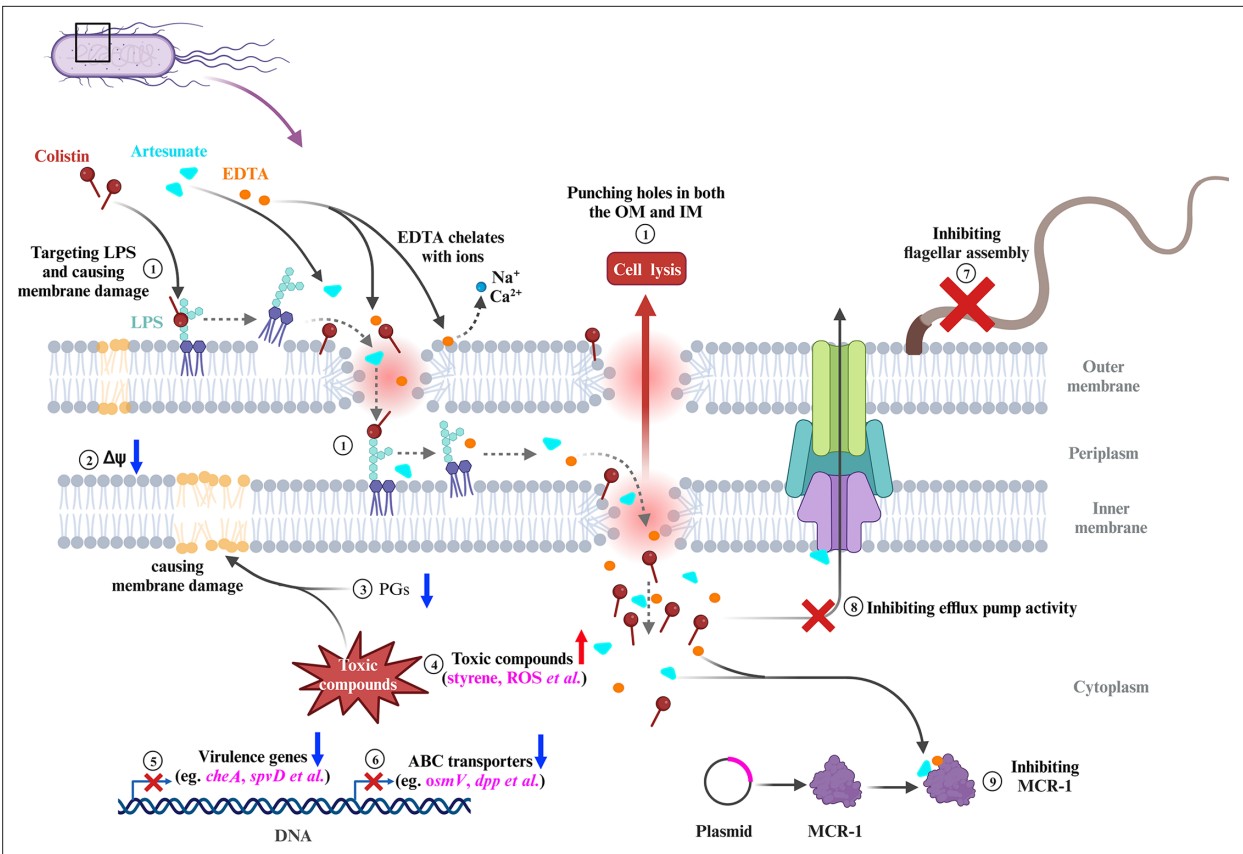

**Figure 8.** Scheme summarizing the proposed mechanisms that artesunate (AS) and EDTA enhance the antibacterial effect of colistin (COL) against *Salmonella*. ① COL and EDTA cause the membrane damage by targeting LPS and chelating cations, which punch holes in both the outer membrane (OM) and inner membrane (IM). ② AEC combination collapses the $\Delta\phi$ component of proton motive force (PMF) in *Salmonella*. ③ The downregulated lipid compounds prostaglandins (PGs) lead to perturbation of membrane phospholipids in cell membranes. ④ Accumulation of toxic compounds (e.g., styrene, ROS) could accelerate the destruction of cell membrane. ⑤ The downregulation of chemotaxis, flagellar assembly, and *Salmonella* infection-related genes indicates impaired virulence of *Salmonella*. ⑥ The downregulation of genes in ABC transporters indicates impaired stress tolerance of *Salmonella*. ⑦ AEC treatment results in the downregulation of flagellar assembly-related genes and defective of flagellum. ⑧ AS was capable of significantly enhancing the antibacterial activities of antibiotics against *E. coli,* via inhibition of the efflux pumps. ⑨ AS could be developed as an MCR-1 inhibitor, which may work synergistically with the EDTA chelation to inhibit MCR-1 and contribute to reverse the COL resistance of *mcr-1*-harboring *Salmonella* strains. FC: fold change.

# Materials and methods

## Bacteria strains and agents

A total of nine bacteria strains were used in this study (*Table 2*), including a multidrug-susceptible standard strain of *Salmonella* Typhimurium CVCC541 (named JS), a COL-susceptible clinical strain of *Salmonella* (named S34), four COL-resistant clinical strains of *Salmonella* (named S16, S20, S13, and S30), a COL-resistant clinical strain of *E. coli* (named E16), and two strains of intrinsically COL-resistant species (*M. morganii* strain M15 and *P. mirabilis* strain P01). COL was purchased from Shengxue Dacheng Pharmaceutical, China. COL and EDTA-2Na were both dispersed into water at a final concentration of 640 mg/L and 10,000 mg/L, respectively. AS was purchased from Meilunbio (Dalian, China) and dissolved in water with 10% (V/V) N, N-dimethylformamide at a final concentration of 5000 mg/L. NPN was purchased from Sigma-Aldrich, USA. PI was purchased from Thermo Fisher Scientific, USA. The 3,3-dipropylthiadicarbocyanine iodide DiSC$_3$(5) and ethidium bromide (EtBr) were purchased from Aladdin, China. BCECF-AM, Reactive Oxygen Species Assay Kit, and Hydrogen Peroxide Assay Kit were purchased from Beyotime, China.

## Antibacterial activity in vitro

### Antimicrobial susceptibility testing

The MICs of COL, AS, and EDTA against all strains were determined by the twofold serial broth microdilution method according to CLSI guidelines (*Wayne, 2021*) The double and triple combination strategies were carried out as follows: AS (1/4, 1/8, or 1/16 MIC of AS) + COL, EDTA (1/4, 1/8, or 1/16 MIC of EDTA) + COL, AS (1/4 MIC of AS) + EDTA (1/4, 1/8, 1/16, or 1/32 MIC of EDTA) + COL, AS (1/8 MIC of AS) + EDTA (1/4, 1/8, 1/16, or 1/32 MIC of EDTA) + COL. These medication strategies, including COL alone, AS + COL, EDTA + COL, AS + EDTA + COL, are abbreviated as C, AC, EC, and AEC, respectively. When determining the MICs of COL after drug combinations, AS or/and EDTA were pre-added into MHB broth with different final concentrations, namely 1/4, 1/8, 1/16 or 1/32 MIC of AS (312.5, 156.25, 78.13, 39.06 mg/L) or EDTA (e.g., 31.25, 15.63, 7.81, 3.91 mg/L for JS, S16, and S30 strains), then COL was added and diluted to make a twofold dilution series. The lowest concentrations with no visible growth of bacteria were defined as MIC values of COL after drug combinations.

### Time-kill assays

Time-kill assays were performed against the *Salmonella* strains JS, S16 (*mcr-1⁻*), and S30 (*mcr-1⁺*) with COL alone as well as in combinations (AC, EC, AEC). When combined with COL (0.1 or 2 mg/L), AS and EDTA were added at final concentrations equivalent to their 1/8 MICs. Overnight cultures were diluted 1:100 in fresh LB medium and grown to an $OD_{600}$ of 0.5, then treated with different combinations for 24 hr. The cultures were serially diluted tenfold and spread over sterile nutrient agar at 0.5, 4, 8, 12, and 24 hr. Bacterial colonies on individual plates were counted after overnight incubation at 37 °C and expressed as the $log_{10}$ of colony-forming units/mL (CFU/mL).

## Fluorescent probe-permeability assays

Overnight cultures of *Salmonella* strains S16 (*mcr-1⁻*) and S30 (*mcr-1⁺*) were diluted 1:100 in fresh LB medium and grown to an $OD_{600}$ of 0.7. Cells were harvested and washed twice with PBS or HEPES, then resuspended in the same buffer to $OD_{600} \approx 0.5$ for further analysis. Different drug combinations or fluorescent probes were added and incubated when necessary. The medication strategies used in the fluorescence probe assays were as follows: C, AC, EC, and AEC. The final concentration of COL was 0.1 or 2 mg/L when used alone or in drug combinations. AS and EDTA were added at final concentrations equivalent to their 1/8 MICs when used in drug combinations. Fluorescence intensity was measured with Spark 10M microplate spectrophotometer (Tecan, Switzerland).

### Cell membrane integrity assay

Bacterial suspensions in HEPES were mixed with either fluorescent probe NPN or PI to a final probe concentration of 10 µM for NPN or 15 µM for PI. After incubation at 37°C for 0.5 hr, bacterial suspensions were then mixed with different drug combinations and incubated for another 1 hr. Fluorescence measurements were then taken with the excitation wavelength at 350 nm (or 535 nm) and emission wavelength at 420 nm (or 615 nm) for NPN (or PI).

### Proton motive force assay

Bacterial suspensions in PBS were incubated with either $DiSC_3(5)$ (0.5 µM) or pH-sensitive fluorescent probe BCECF-AM (20 µM) for 0.5 hr to determine the membrane potential ($\Delta\psi$) and pH gradient ($\Delta pH$). Then these suspensions were mixed with different drug combinations and incubated for another 1 hr. Finally, the fluorescence was measured with the excitation wavelength of 622 nm (or 488 nm) and emission wavelength of 670 nm (or 535 nm) for $DiSC_3(5)$ (or BCECF-AM).

### Total ROS measurement

The ROS-sensitive fluorescence indicator 2', 7'-dichlorodihydro-fluorescein diacetate (DCFH-DA, 10 µM) was used to assess the ROS levels in bacterial cells. Bacterial suspensions in PBS were mixed with DCFH-DA and incubated for 0.5 hr, then different drug combinations were added and incubated for another 1 hr and 6 hr. Finally, fluorescence intensity was measured at an excitation wavelength of 488 nm and an emission wavelength of 525 nm.

### Efflux pump assay

Bacterial suspensions in PBS were incubated with ethidium bromide (EtBr) for 0.5 hr in a final concentration of 5 μM. Then different drug combinations were added and incubated for another 1 hr, and the accumulation of EtBr in the cells was evaluated with excitation wavelength of 530 nm and barrier filter of 600 nm.

### H$_2$O$_2$ assay

The bacterial culture conditions and sample preparation were the same as the fluorescent probe-permeability assays section. The luminescence absorbency was measured by a Spapk 10M Microplate reader (Tecan). The cellular H$_2$O$_2$ levels were assessed according to the kit procedure by using a Hydrogen Peroxide Assay Kit. Cell samples were prepared as described above and incubated with different drug combinations for 1.5 hr. Cell precipitates were collected by centrifugation (10,000 rpm), and 200 μL lysis solution were added under gentle shaking. Supernatants were then taken for luminescence (absorbance) detection at 560 nm.

### Scanning electron microscope

Overnight S16 (mcr-1$^-$) culture was diluted 1:100 in fresh LB medium and grown to an OD$_{600}$ of 0.7. Cells were then divided equally and different drug combinations were added, same as that in the fluorescent probe-permeability assays section. After 6 hr incubation at 37°C, cells were washed three times with PBS and fixed with 2.5% glutaraldehyde at 4°C for 24 hr. Samples were then stained in 1% osmium tetroxide and dehydrated in a series of increasing ethanol concentrations (30–100%). The processed samples were dried in Critical Point Dryer (Quorum K850) and sputter-coated with gold. Finally, samples were observed and images were taken with SEM (HITACHI, SU8100).

### Omics analysis

Overnight cultures of S16 (mcr-1$^-$) and S30 (mcr-1$^+$) were diluted 1:100 in fresh LB medium and grown to an OD$_{600}$ of 0.5. Cells were divided equally and different drug combinations were used, including C, AC, EC, and AEC. When combined with COL (2 mg/L), AS and EDTA were added at final concentrations equivalent to their 1/8 MICs. Samples were continually incubated 6 hr, then washed three times with PBS and fast-frozen in liquid nitrogen for further use. Transcriptome and metabolome analysis was performed among different comparison groups, including AC vs. C, EC vs. C, AEC vs. AC, and AEC vs. EC, and carried out by Novogene Co. Ltd (Beijing, China).

#### Transcriptome analysis

The clean reads were mapped to the *Salmonella* Typhimurium DT2 genome (HG326213.1) from NCBI using Bowtie2. The SDEGs were screened with a p-value≤0.05 and |log$_2$Fold Change|≥1. ClusterProfiler software was used to analyze the Gene Ontology functional enrichment (GO, https://geneontology.org/) or Kyoto Encyclopedia of Genes and Genomes pathway enrichment (KEGG, https://www.kegg.jp/kegg/pathway.html) of SDEGs.

#### Metabolome analysis

In this study, samples were analyzed by the non-targeted metabolomics with liquid chromatography-tandem mass spectrometry in either positive ion or negative ion mode. Compound Discoverer 3.1 software (Thermo Scientific, Waltham, MA) was used for the identification of metabolites based on the exact masses and fragmentation spectra. The significant differential metabolites (SDMs) were identified with VIP ≥ 1.0, fold change ≥ 1.2, or fold change ≤ 0.833, p-value<0.05. SDMs were then annotated and classified by KEGG database (https://www.genome.jp/kegg/pathway.html), Human Metabolome Database (HMDB, https://hmdb.ca/metabolites), and Lipidmaps Database (https://www.lipidmaps.org/).

## In vivo antibacterial activity

### Bacterial preparation

Overnight culture of S30 (*mcr-1*+) was diluted 1:100 in fresh LB medium and grown to an $OD_{600}$ of 0.7. Cells were harvested and washed twice with PBS, then the concentration of bacterial suspensions to $1.31 \times 10^6$ CFU/mL for further use.

### Animals and treatments

A total of 36 SPF Kunming mice (6–8-week-old, 18–22 g, half male and half female) were purchased from the Huaxing Experimental Animal Center of Zhengzhou (Zhengzhou, China) and divided into six groups (n=6 per group): (1) PBS control group; (2) COL (10 mg/kg) group; (3) AS (15 mg/kg) group; (4) AS (15 mg/kg) + COL (10 mg/kg) group; (5) EDTA (50 mg/kg) + COL (10 mg/kg) group; and (6) AS (15 mg/kg) + EDTA (50 mg/kg) + COL (10 mg/kg) group. Each mouse was intraperitoneally injected with 100 µL bacterial solution ($1.31 \times 10^5$ CFU). Treatment was initiated at 2 hr post infection and continued for 3 days. Treatment was administered once per day by intraperitoneal injection according to the therapeutic dose mentioned above. Mice were all euthanized, and the spleen and liver of aseptic were collected, weighed, and homogenized with PBS. Then the tissue homogenates were serially diluted with PBS in an appropriate amount, and 100 µL of each dilution was withdrawn and uniformly spread on SS agar plates. Bacterial counts were computed and presented as the mean ± SD $\log_{10}$ CFU/mL after incubated at 37°C for 16–18 hr. Mice were maintained in a barrier facility and guaranteed strict compliance with the regulations of the Administration of Affairs Concerning Experimental Animals approved by the State Council of People's Republic of China (11-14-1988).The mouse experiments were approved by the Henan Science and Technology Department (protocol number SCXK 2019-0002).

## Overexpression of SDEGs in S16 strain

The complete open-reading frame of SDEGs (including *cheA*, *cheY*, *STMDT2-34621*, *aer*, *fliD*, *fliT*, *opuBB*, *gltI*, *dppB*, and *spvD*) was amplified by PCR from the genomic DNA of strain S16. Then these genes were inserted into the multiple cloning site of vector pBAD (Ampicillin+), and the recombinant plasmids were chemically transformed into S16 cells. Finally, the target genes were induced by 0.2% L-arabinose (L-Ara) when necessary.

## RT-PCR analysis

The *mcr-1*+ *Salmonella* S30 strain was grown to $OD_{600} \approx 0.5$, then incubated with double and triple combination strategies (same as the time-kill assays section) for 6 hr. Total RNA was extracted and isolated by following the phenol-chloroform method, and then reverse transcribed to cDNA using PrimeScriptRT reagent Kit (TaKaRa). Primers for RT-PCR were designed according to *Yi et al., 2022*, listed in *Supplementary file 1a*. The 16S rRNA gene was chosen as a housekeeping gene. The relative expression ratio of the gene tested was determined using the $2^{-\Delta (\Delta CT)}$ method compared to that of AS and EDTA treatment.

## Molecular docking

Crystal structure of the C-terminal catalytic domain of MCR-1 with two zinc ions (PDB ID: 5GRR) was obtained from the Protein Data Bank (PDB, http://www.rcsb.org). 3D structure of AS was downloaded from The Pubchem Project (PubChem CID: 6917864). The region containing the previously reported active site within MCR-1 was defined as the binding site for docking simulations. AutoDock software was used for the flexible ligand docking between MCR-1 and AS. The results that were ranked permitted to energy values of the docking (kcal/mol), and the lower the value, the more likely the ligand-active site bind. The Discovery Studio molecular graphics system was used to preliminarily estimate and further confirm the modes of interaction with binding site residues.

## Competitive inhibitory assays

Two polypeptides from MCR-1 (named $P_u$ and $P_m$) were synthesized by Sangon Biotech Company. $P_u$ (5'-CG**TS**TA**Y**SVP-3') and $P_m$ (5'-CG**AA**TA**A**SVP-3') were unmutated and mutated peptides from the binding sites THR283, SER284, and TYR287, respectively. AS and AS + EDTA were pre-incubated

with $P_u$ or $P_m$ for 2 hr, and then added into the *mcr-1*[+] *Salmonella* S30 strain. The MICs of COL were determined by the twofold serial broth microdilution method according to CLSI guidelines (*Wayne, 2021*).

## Motility assays

Motility assays were performed using 0.3% agar plates containing AS, EDTA, COL alone or drug combinations. The final concentration of COL was 2 mg/L when used alone or in drug combinations. AS and EDTA were added at final concentrations equivalent to their 1/8 MICs when used in drug combinations. Overnight culture of S16 (*mcr-1*[-]) and S30 (*mcr-1*[+]) was diluted 1:100 in fresh LB medium and grown to an $OD_{600}$ of 0.5, and inoculated on 0.3% agar plates. The migration distance (cm) was measured and recorded for 48 hr at 37°C.

## Statistical analysis

Statistical analysis was conducted using GraphPad Prism 9 and SPSS software. All data were derived from $n \geq 3$ biological replicates and presented as mean ± SD. Without specific indication, differences between the independent groups (*$p < 0.001$) were assessed with Student's *t*-test or one-way ANOVA.

## Acknowledgements

This work was supported by the National Natural Science Foundation of China (nos. 32102716 and 32373069). This study was carried out in accordance with the guidelines of the Henan Agricultural University Animal Ethics Committee.

## Additional information

### Funding

| Funder | Grant reference number | Author |
| --- | --- | --- |
| National Natural Science Foundation of China | 32102716 | Yajun Zhai |
| National Natural Science Foundation of China | 32373069 | Gongzheng Hu |

The funders had no role in study design, data collection and interpretation, or the decision to submit the work for publication.

### Author contributions

Yajun Zhai, Conceptualization, Supervision, Funding acquisition, Project administration, Writing – review and editing; Peiyi Liu, Xueqin Hu, Changjian Fan, Data curation, Investigation, Methodology; Xiaodie Cui, Qibiao He, Writing - original draft; Dandan He, Analyzed the results; Xiaoyuan Ma, Analyzed the results; Gongzheng Hu, Data curation, Supervision, Funding acquisition, Project administration, Writing – review and editing

### Author ORCIDs

Yajun Zhai https://orcid.org/0009-0002-2218-4819

### Ethics

Mice were maintained in a barrier facility and guaranteed strict compliance with the regulations for the Administration of Affairs Concerning Experimental Animals approved by the State Council of People's Republic of China (11-14-1988). The mouse experiments were approved by the Henan Science and Technology Department (protocol number SCXK 2019-0002).

Reviewer #1 (Public review): https://doi.org/10.7554/eLife.99130.3.sa1
Reviewer #2 (Public review): https://doi.org/10.7554/eLife.99130.3.sa2
Author response https://doi.org/10.7554/eLife.99130.3.sa3

# Additional files

## Supplementary files
Supplementary file 1. Supplementary tables. (**a**) Sequences of primers used in this study. (**b**) The antibacterial activities of COL, AS, and EDTA against the tested strains after single and double combinations. (**c**) The antibacterial activities of COL against the tested strains after single and triple combinations. (**d**) The MICs of different antimicrobial drugs against the S16 and S30.

MDAR checklist

## Data availability
Data have been made available freely online. Transcriptome data have been submitted to the Sequence Read Archive database (SRA, https://www.ncbi.nlm.nih.gov/sra) under the BioProject accession number PRJNA1036120 (S16 strain) and PRJNA1036408 (S30 strain). Metabolome data have been submitted to the MetaboLights database (https://www.ebi.ac.uk/metabolights) under accession number MTBLS8875.

The following datasets were generated:

| Author(s) | Year | Dataset title | Dataset URL | Database and Identifier |
|---|---|---|---|---|
| Yajun Z, Peiyi L, Xueqin H, Changjian F, Xiaodie C, Qibiao H, Dandan H, Xiaoyuan M, Gongzheng H | 2024 | RNA-Seq of *Salmonella* | https://www.ncbi.nlm.nih.gov/sra/?term=PRJNA1036120 | NCBI Sequence Read Archive, PRJNA1036120 |
| Yajun Z, Peiyi L, Xueqin H, Changjian F, Xiaodie C, Qibiao H, Dandan H, Xiaoyuan M, Gongzheng H | 2024 | RNA-Seq of *Salmonella* | https://www.ncbi.nlm.nih.gov/sra/?term=PRJNA1036408 | NCBI Sequence Read Archive, PRJNA1036408 |
| Yajun Z, Peiyi L, Xueqin H, Changjian F, Xiaodie C, Qibiao H, Dandan H, Xiaoyuan M, Gongzheng H | 2024 | Artesunate, EDTA and colistin work synergistically against MCR-negative and -positive colistin-resistant *Salmonella* | https://www.ebi.ac.uk/metabolights/editor/MTBLS8875/descriptors | MetaboLights, MTBLS8875 |

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
