## [Editor Report · eLife Assessment]

This **valuable** study addresses the growing threat of multidrug-resistant (MDR) pathogens by focusing on the enhanced efficacy of colistin when combined with artesunate and EDTA against colistin-resistant *Salmonella* strains. The evidence is **solid**, supported by comprehensive microbiological assays, molecular analyses, and in vivo experiments demonstrating the effectiveness of this synergic combination.

---

## [Referee Report · Reviewer #1 (Public review)]

Summary:

The study addresses the growing threat of multi-drug-resistant (MDR) pathogens, focusing on the efficacy of colistin (COL), a last-resort antibiotic, and its enhanced activity when combined with artesunate (AS) and ethylenediaminetetraacetic acid (EDTA) against colistin-resistant *Salmonella* strains. The researchers aim to explore whether these combinations can restore the effectiveness of colistin and understand the underlying mechanisms. The study used a combination of microbiological and molecular techniques to evaluate the antibacterial activity and mechanisms of action of COL, AS, and EDTA. Key methods included: (i) Antimicrobial Susceptibility Testing: Determining minimum inhibitory concentrations (MICs) of COL, AS, and EDTA, both alone and in combination, against various *Salmonella* strains; (ii) Time-Kill Assays: Measuring bacterial growth inhibition over time with different drug combinations; (iii) Fluorescent Probe-Permeability Assays: Assessing cell membrane integrity using fluorescent dyes; (iv) Proton Motive Force Assay: Evaluating the impact on the electrochemical proton gradient (PMF); (v) Reactive Oxygen Species (ROS) Measurement: Quantifying intracellular ROS levels; (vi) Scanning Electron Microscopy (SEM): Observing morphological changes in bacterial cells; and (vii) Omics Analysis: Transcriptome and metabolome profiling to identify differentially expressed genes (DEGs) and significant differential metabolites (SDMs). The combination of COL, AS, and EDTA (AEC) showed significant antibacterial activity against colistin-resistant *Salmonella* strains, reducing the MICs and enhancing bacterial killing compared to individual treatments. The AEC treatment caused extensive damage to both the outer and inner bacterial membranes, as evidenced by increased fluorescence of membrane-impermeant dyes and SEM images showing deformed cell membranes. AEC treatment selectively collapsed the Δψ component of PMF, indicating disruption of vital cellular processes. The combination therapy increased intracellular ROS levels, contributing to bacterial killing. Transcriptome data revealed changes in genes related to two-component systems, flagellar assembly, and ABC transporters. Metabolome analysis highlighted disruptions in pathways such as arachidonic acid metabolism. The findings suggest that AS and EDTA can potentiate the antibacterial effects of colistin by disrupting bacterial membranes, collapsing PMF, and increasing ROS levels. This combination therapy could serve as a promising approach to combat colistin-resistant *Salmonella* infections.

Strengths:

- The study employs a wide range of techniques to thoroughly investigate the antibacterial mechanisms and efficacy of the drug combinations.

- The results are consistent across multiple assays and supported by both in vitro and in vivo data.

- Combining AS and EDTA with COL represents a novel strategy to tackle antibiotic resistance.

Weaknesses:

- The methodology used for interpreting and reporting time-kill assay results.

Comments on revised version:

Overall, the authors have adequately addressed the suggestions provided.

---

## [Referee Report · Reviewer #2 (Public review)]

The study by Zhai et al describes repurposing of artesunate, to be used in combination with EDTA to resensitize *Salmonella* spp. to colistin. The observed effect applied both to strains with and without mobile colistin resistance determinants (MCR). It is known since earlier that EDTA in combination with colistin has an inhibitory effect on MCR-enzymes, but at the same time both colistin and EDTA can contribute to nephrotoxicity, something which is also true for artesunate. Thus, the triple combination of three nephrotoxic agents has significant challenges in vivo, which is not particularly discussed in this paper.

The study is sound from a methodological point of view and has many interesting angles to address mechanistically how the three compounds can synergize.

Comments on revised version:

After having read the revised version, I have the following comments:

(1) The antimicrobials tested in Figure 9 are not really very relevant. I would want to see carbapenems and novel beta-lactam/beta-lactamase inhibitors rather than many old drugs with a debatable role in the treatment of Gram-negative infections. At least the authors should be able to test carbapenem resistance

(2) The genomics analysis of the strains should be fairly quick - both in terms of characterizing the mobile resistome and the sequence types. There are publicly available databases for this purpose

The rest of my comments have been addressed in the revised version. There are still some remaining valid points from other reviewers that could be debatable whether they should be address. The authors refer to plans of studying these aspects in subsequent studies, but it could be discussed whether some of the data could be expected already in this study.

---

## [Author Response]

The following is the authors’ response to the original reviews.

**eLife Assessment:**
This study provides valuable insights, addressing the growing threat of multi-drug-resistant (MDR) pathogens by focusing on the enhanced efficacy of colistin when combined with artesunate and EDTA against colistin-resistant *Salmonella* strains. The evidence is solid, supported by comprehensive microbiological assays, molecular analyses, and in vivo experiments demonstrating the effectiveness of this synergic combination. However, the discussion on the clinical application challenges of this triple combination is incomplete, and it would benefit from addressing the high risk associated with using three potential nephrotoxic agents in vivo.

The development of novel pharmaceutical dosage forms, pharmacokinetic, pharmacodynamic and safety analysis of the triple combination will be further conducted in our next study to provide a theoretical basis for the next clinical drug use. The discussion of potential toxicity of AS, colistin, EDTA and the triple combination have been added in line 318 to 337.

**Public Reviews:**

**Reviewer #1 (Public Review):**
(1) The study focuses on a limited number of *Salmonella* strains, and broader testing on various MDR pathogens would strengthen the findings.

The number of COL-resistant clinical strains that actually used was larger than that mentioned in our original article, when evaluating the antimicrobial activities of AS, EDTA, COL alone or drug combinations. But, considering that there were superfluous results of *mcr-1* positive *Salmonella* strains, we omitted these results (Table supplement 7 and 8 in revised supplement materials) to avoid redundant data presentation in the original article. Additionally, much more gram-negative and -positive MDR bacteria, such as *Klebsiella pneumoniae*, *Pseudomonas aeruginosa* and *Staphylococcus aureus* will be selected for the next study including the development of novel pharmaceutical dosage forms, pharmacokinetic, pharmacodynamic and safety analysis et al.

(2) While the study elucidates several mechanisms, further molecular details could provide deeper insights into the interactions between these drugs and bacterial targets.

In our next study, further molecular details will be focused on the regulatory targets of CheA and SpvD-related pathways, as well as the precise inhibition targets of MCR protein by the triple combination, through the generation of deletion or point mutations, and analysis of intermolecular interactions.

(3) The time-kill experiment was conducted over 12 hours instead of the recommended 24 hours. To demonstrate a synergistic effect among the drugs, a reduction of at least 2 log10 in colony count should be shown in a 24-hour experiment. Additionally, clarifying the criteria for selecting drug concentrations is important to improve the interpretation of the results.

The time-kill experiment of 24 hours have been re-executed and could be used to replace the Figure 1 in the original paper. The New Figure 1 has been uploaded and the change do not affect our interpretation of the result.

Although in vitro studies have determined that with increasing dose of AS and EDTA, the antibacterial synergistic activity was gradually enhanced, and meanwhie, may also resulting in more toxic side effects. Thus, in our study, the 1/8 MICs of AS and EDTA were selected to ensure excellent antibacterial activity whereas minimize the potential toxicity. The instructions on the selection of drug concentration have been added in line 323 to 326.

(4) While the combination of EDTA, artesunate, and colistin shows promising in vitro results against *Salmonella* strains, the clinical application of this combination warrants careful consideration due to potential toxicity issues associated with these compounds.

The development of novel pharmaceutical dosage forms, pharmacokinetic, pharmacodynamic and safety analysis of the triple combination will be further conducted in our next study to provide a theoretical basis for the next clinical drug use.

**Reviewer #2 (Public Review):**
(1) The study by Zhai et al describes repurposing of artesunate, to be used in combination with EDTA to resensitize *Salmonella* spp. to colistin. The observed effect applied both to strains with and without mobile colistin resistance determinants (MCR). It was already known that EDTA in combination with colistin has an inhibitory effect on MCR-enzymes, but at the same time, both colistin and EDTA can contribute to nephrotoxicity, something which is also true for artesunate. Thus, the triple combination of three nephrotoxic agents has significant challenges in vivo, which is not particularly discussed in this paper.

The discussion of potential toxicity of triple combination has been added in line 318 to 337.

(2) The selection of strains is not very clear. Nothing is known about the sequence types of the strains or how representative they are for strains circulating in general. Thus, it is difficult to generalize from this limited number of isolates, although the studies done in these isolates are comprehensive.

The tested strains in this study were all COL-resistant clinical isolates, and the genome sequencing and comparative analysis of these strains have not been analyzed. The antibacterial activities of different antimicrobial drugs against the S16 and S30 strains have been measured and listed in the Table supplement 9 within revised supplement materials. Considering that the number of COL-resistant clinical strains that actually used was larger than that mentioned in our original article (see the NO.1 response to the Public Reviewer #1), we think that the results obtained in this study could be representative to some extent.

(3) Nothing is known about the susceptibility of the strains to other novel antimicrobial agents. Colistin has a limited role in the treatment of gram-negative infections, and although it can be used sometimes in combination, it is not clear why it would be combined with two other nephrotoxic agents and how this could have relevance in a clinical setting.

The antibacterial activities of different antimicrobial drugs against the S16 and S30 strains have been measured and listed in the Table supplement 9 within revised supplement materials. Additionally, the discussion of potential toxicity of triple combination has been added in line 318 to 337.

(4) It is not clear whether their transcriptomics analysis should at least be carried out in duplicate for reasons of being able to assess reproducibility. It is also not clear why the samples were incubated for 6 hours - no discussion is presented on the selection of a time point for this.

As it can be seen from the time kill curves that the survival number of bacteria started to decrease after 4 h incubation of drug combinations. If the incubation time is too short (for example less than 4 h), the differentially expressed genes can not be fully revealed, while too long incubation time (such as 8 h and 12 h) may lead to a significant CFU reduction of bacteria, and result in inaccurate sequencing results. Therefore, we selected the incubation time 6 h, at which point drugs exhibited significant antibacterial effects and there were also enough survival bacteria in the sample for transcriptome analysis. Each sample had three replications to preserve the accuracy of results.

(5) Discussion is lacking on the reproducibility and selection of details for the methodology.

The results obtained in this paper have been repeated several times, which indicated that the detailed operation steps described in the materials and methods section were reproducibility. To avoid redundancy, we did not include too much details in the discussion section.

**Reviewer #3 (Public Review):**
(1) Number of strains tested.

The number of COL-resistant clinical strains that actually used was larger than that mentioned in our original article (see the NO.1 response to the Public Reviewer #1)

(2) Response to comment: Lack of data on cytotoxicity.

The pharmacokinetic, pharmacodynamic and safety analysis of the triple combination will be further conducted in our next study to provide a theoretical basis for the next clinical drug use.

**Recommendations For The Authors:**

**Reviewer #1 (Recommendations For The Authors):**
(1) Introduction:The introduction should provide more context about the pathogen *Salmonella*, its significance in both human and veterinary medicine, and the impact of colistin resistance in these pathogens. *Salmonella* is a leading cause of foodborne illnesses worldwide, resulting in substantial morbidity and mortality. It can cause a range of diseases, from gastroenteritis to more severe systemic infections like typhoid fever and invasive non-typhoidal salmonellosis. In veterinary medicine, Salmonella infections can lead to significant economic losses in livestock industries due to illness and death among animals, as well as through the contamination of animal products.

The description has been added in the introduction section in line 47 to 53.

(2) Results and Discussion:(1) While the combination of EDTA, artesunate, and colistin shows promising in vitro results against *Salmonella*, the clinical application of this combination warrants careful consideration due to potential toxicity issues associated with these compounds. Colistin is known for nephrotoxicity and neurotoxicity, limiting its use to severe cases where the benefits outweigh the risks. EDTA, as a chelating agent, can disrupt essential metal ions in the body, posing risks of metabolic imbalances. Although it has clinical applications, primarily in cases of heavy metal poisoning, its use as an adjuvant in antibiotics may present risks. Although generally well-tolerated for malaria, interactions of artesunate with other drugs and long-term safety in combined therapies require thorough investigation.

The discussion of potential toxicity of triple combination has been added in line 318 to 337.

(2) Table 1: The manuscript mentions that some strains used in the study are mcr-positive and mcr-negative. It is important to indicate in Table 1, in addition to the identification of *Salmonella* species, which strains are mcr-positive or mcr-negative.

The relevant information has been added in Table 1.

(3) Figure 2: What is the authors' hypothesis regarding the growth curves labeled "a" and "e" where strains JS and S16 resume growth 12 hours after treatment with AS? In the legend of Figure 2, describe what was used as the "positive control group."

The growth curves labeled “a” and “e” were in Figure 1. After incubated with AC for 8 h, the survival CFUs of JS and S16 strains showed a slightly reduction, but there were still living cells. Since the bactericidal activity of AC is not strong enough to exert sustained bactericidal activity, these remaining living cells will resume growth after treatment with AC for 12 h. The “positive control group” in the legend of Figure 2 has been indicated in line 724.

(4) What is the authors' hypothesis for the differences observed in the transcriptome and metabolome?

The changes in gene transcription level may cause corresponding changes in protein level, but these proteins are not all involved in the bacterial metabolic process. For example, MCR protein is encoded by the COL resistance related gene *mcr*, which mediates the modification of lipid A, but are not involved in the cellular metabolic process. Therefore, the transcriptome change of *mcr* gene may affect the protein production of MCR, nor the bacterial metabolic processes, so there are differences observed in the transcriptome and metabolome.

(5) In some parts of the text, the authors state that artesunate and EDTA potentiate the action of colistin, which is a bacteriostatic drug. However, in other parts, the authors describe the effect of the AEC combination as bacteriostatic (Abstract: line 32; Results: line 179). How do the authors explain this inconsistency?

The artesunate and EDTA could be regarded as “adjuvants” for the bacteriostatic drug colistin. Adjuvants itself act no or weak antibacterial effect on bacteria. For antimicrobial drugs, the “adjuvants” are compounds that generally used in combination with antibacterial drugs to re-sensitizing bacteria that have developed drug resistance. Thus, in this paper the AEC combination could be regared as bacteriostatic.

(6) According to Brennan & Kirby (2019; doi: 10.1016/j.cll.2019.04.002), to evaluate the synergism between different drug combinations, bacterial growth curves need to be assessed over 24 hours. If the colony count is {greater than or equal to} 2 log10 lower than that of the most active antimicrobial alone, the combination is considered synergistic. Based on the growth curve results shown in Figure 1, the experiment was conducted for 12 hours, and in some cases, only a small reduction in growth was observed, even at the maximum concentration of colistin. Moreover, in some cases, the curve resumes rising between 8 and 12 hours. What is the authors' hypothesis in this case? It is important to conduct the assay over 24 hours to confirm the synergism between these drugs.

The time-kill experiment of 24 hours have been re-executed and could be used to replace the Figure 1 in the original paper. Additionally, the phenomenon that “the curve resumes rising between 8 and 12 hours” has been explained in the response to comment of “Reviewer #1 (Recommendations For The Authors), Results and Discussion, (3) Figure 2”.

(7) To prove that CheA and SpvD play a critical role in the effect of the AEC combination, deletion of these genes should be performed, and the mutant strains should be tested.

The deletion of *cheA* and *spvD* will be carried out in our next study.

(8) To demonstrate that the flagellum is no longer assembled, a transmission electron microscopy image using antibodies against flagellin should be performed, along with motility tests.

The motility assays have been performed and displayed as Figure supplement 5 in the revised supplement materials.

(9) Figure 7: In the X-axis legend, specify what "model" refers to.

The “model” refers to the PBS control group that mice were treated with PBS after the intraperitoneal injection of 100 µL bacterial solution (1.31 × 10^5^ CFU).

(10) Figure 8 Legend: In the legend of Figure 8 (line 717), are the authors referring to *E. coli* or *Salmonella*?

It referred to *Salmonella*, which has already been illustrated in the headline of Figure 8 in the revised manuscript.

(3) Materials and Methods:(1) Bacterial Strains and Agents: It would be beneficial to include in the table the species of the strains used in the study, as well as the concentrations of colistin, artesunate, and EDTA utilized (lines 321 - 332).

We have ever tried to add the above information to Table 1, but the addition of this information would make the table too large and beyond the margins, which is not conducive to the layout design of the table, so we chose to display these information in the materials and methods section instead of the table.

(2) Antibacterial Activity In Vitro: Ensure clarity and well-defined ranges for the concentrations of colistin, EDTA, and artesunate used separately and in combinations (lines 335 - 344).

The drug concentrations have been listed in line 369 to 371.

(3) Time-Kill Assays: Clarify the criteria for selecting concentrations, whether based on MICs or peak and trough concentrations relevant to human and animal treatments with colistin (lines 345 - 351).

Although in vitro studies have determined that with increasing dose of AS and EDTA, the antibacterial synergistic activity was gradually enhanced, and meanwhie, may also resulting in more toxic side effects. Thus, in our study, the 1/8 MICs of AS and EDTA were selected to ensure excellent antibacterial activity whereas minimize the potential toxicity. The instructions on the selection of drug concentration have been added in line 323 to 326.

(4) General Corrections: Throughout the manuscript, correct typographical errors and consistently include the concentration values in mg/L alongside the MIC fractions. Specify the strains used for all experiments to ensure clarity. In the manuscript, the term "medication regimens" is used to describe the experimental setups involving different combinations of drugs tested in vitro. To improve accuracy and clarity, it is recommended to use the term "drug combination" instead. This term is more appropriate for in vitro experiments and will help avoid confusion with clinical treatment protocols.

The typographical errors have been checked and corrected throughout the manuscript, and the “medication regimens” have been replaced by “drug combinations”.

**Reviewer #2 (Recommendations For The Authors):**
Please see above for recommendations on what can be done to improve the manuscript.

While other omics analyses have been conducted herein, the authors do not comment on the genomic analysis of their own strains. It would have been a natural step to sequence all the strains used in the experiments.

Due to limited program funding, the genome sequencing and comparative analysis of these strains have not been analyzed. The antibacterial activities of different antimicrobial drugs against the S16 and S30 strains have been measured and listed in the Table supplement 9 within revised supplement materials.

Some minor comments:(1) There are some spelling errors - e.g. "bacteria strains" instead of "bacterial strains".

The grammar and spelling errors have been corrected throughout the manuscript.

(2) I would avoid words like "unfortunately".

The word “unfortunately” has been changed.

(3) Some MIC-values in Table 1 seem incorrect - e.g. 24 mg/L. This is not a 2-log value - the value should be 32 mg/L if the dilution series has been carried out correctly.

We are so sorry for the mistake. The data has been corrected, and we also checked other data.

**Reviewer #3 (Recommendations For The Authors):**
Below are some suggestions.(1) Sentences L47 & L48 "Infections with antibiotic-resistant pathogens, especially carbapenemase-producing Enterobacteriaceae, represent an impending catastrophe of a return to the pre-antibiotic era" - this is slightly exaggerated! I also wonder if we need to use Enterobacterales instead of Enterobacteriaceae.

The sentences in L47 & L48 have been changed. We googled the “carbapenemase-producing Enterobacteriaceae” and found it is a high-frequency word in numerous reports.

(2) L48. The drying up of the antibiotic discovery pipeline is NOT necessarily the reason to use colistin as a drug of last resort!

The sentence has been revised.

(3) The manuscript requires extensive English editing but has merit based on the strong compilation of data.

We have optimized and revised the writing of the whole article.

(4) I suggest the authors have some data on the cytotoxicity of AS alone, colistin alone, and both of them against eucaryotic cells (Caco-) and if possible determine IS (index selectivity). This additional experiment will strengthen the quality of the manuscript. The authors must also explain how to put such tri-therapy into practice.

The development of novel pharmaceutical dosage forms, pharmacokinetic, pharmacodynamic and safety analysis of the triple combination will be further conducted in our next study to provide a theoretical basis for the next clinical drug use. The discussion of potential toxicity of AS, colistin, EDTA and the triple combination have been added in line 318 to 337.